# IN-DEPTH ROBUSTNESS ANALYSIS FOR VISION-LANGUAGE-ACTION MODELS

## ABSTRACT

Visual–Language–Action (VLA) models report impressive success rates on robotic manipulation benchmarks, yet these results may mask fundamental weaknesses in robustness. We perform a systematic vulnerability analysis by introducing controlled perturbations across seven dimensions: objects layout, camera viewpoints, robot initial states, language instructions, light conditions, background textures and sensor noise. We comprehensively analyzed multiple state-of-the-art models and revealed consistent brittleness beneath apparent competence. Our analysis exposes critical weaknesses: models exhibit extreme sensitivity to perturbation factors including camera viewpoints and robot initial states, with performance dropping from 95% to below 30% under modest perturbations. Surprisingly, models are largely insensitive to language variations, with further experiments revealing that models tend to ignore language instructions completely. Our findings challenge the assumption that high benchmark scores equate to true competency and highlight the need for evaluation practices that assess reliability under realistic variation.

## 1 INTRODUCTION

Recent advances in Visual–Language–Action (VLA) models have led to impressive performance on standardized benchmarks, with many systems achieving near-perfect success rates on tasks in controlled simulation environments (Kim et al., 2024; 2025; Li et al., 2025; Black et al.; Pertsch et al., 2025; Hung et al., 2025; Cen et al., 2025; Tan et al., 2025). However, these headline numbers often conceal critical deficiencies in the underlying models. In fact, a closer inspection reveals that contemporary VLA systems tend to exhibit a fragile robustness, struggling to maintain performance when faced with even minor variations in environmental conditions or task parameters.

The prevailing evaluation methodologies (Liu et al., 2023; Li et al., 2024c) focus on aggregate success rates under static, ideal conditions. While such metrics provide valuable baselines for comparing different approaches, they fail to capture the stability and reliability of learned policies under realistic variations. This approach tends to obscure the models' inability to handle subtle variations that are intrinsic to any realistic task setting(Wang et al., 2025; Müller, 2019; Zhang et al., 2024)—even if those tasks remain within the realm of simulation. For example, models trained to excel under fixed camera angles or consistent illumination often fail to generalize when confronted with slight shifts in viewpoint or minor changes in the robot's initial configuration. This gap is especially problematic for VLA models, which must integrate information across multiple modalities and maintain coherent behavior despite perturbations in any of these input channels.

To uncover these hidden vulnerabilities, we conduct a comprehensive analysis of contemporary VLA models using the LIBERO (Liu et al., 2023) benchmark as a diagnostic tool. By systematically varying key factors such as camera viewpoints, robot initial states, language instructions, light conditions, background textures, sensor noise, and object layout, we expose the brittle nature of these models. Our analysis shows that even nominal modifications can lead to steep drops in performance. This indicates that, rather than achieving true multimodal understanding, current VLA architectures rely on overfitting to specific, narrowly defined cues provided during training.

Our study highlights several core weaknesses in contemporary VLA models: **Vulnerability to Visual Shifts:** an over-reliance on fixed visual features leads to failure under variations in camera angle or illumination; **Inadequate Kinematic Reasoning:** limited generalization across different

initial robot configurations reflects a lack of deep kinematic understanding; **Superficial Language Interaction:** linguistic inputs are often underutilized or even completely ignored, as shown by the minimal impact of instruction variation.

Through this work, we provide:

1. A detailed vulnerability analysis of current VLA models through systematic parameter variation.

2. A diagnostic framework for identifying and quantifying the impact of perturbations on model performance.

3. Critical insights into the mismatch between apparent multimodal competence and actual robust understanding.

Our findings challenge the assumption that high benchmark scores equate to true competency, urging the community to re-evaluate current evaluation practices and focus on building models that are robust in the face of inherent variability. This work is a step toward developing VLA systems that are not only high-performing but also genuinely reliable and adaptable.

## 2 How Do Single-Dimension Perturbations Affect VLA Models?

### 2.1 Perturbation Factors

We systematically evaluate how different perturbation factors affect VLA performance and study seven common single-dimension perturbations applied to the evaluation episodes: (1) *Objects Layout*: add confounding objects and/or shift the target object's position. (2) *Camera Viewpoints*: change the viewpoint/pose and field-of-view of the third-person camera. (3) *Robot Initial States*: change the manipulator's initial pose. (4) *Language Instructions*: rewrite task instructions to increase linguistic richness and complexity. (5) *Light Conditions*: vary illumination intensity, direction, color, and shadow patterns. (6) *Background Textures*: modify table/scene textures and materials. (7) *Sensor Noise*: inject photometric distortions (e.g., jitter, Gaussian blur) into input images. Full per-factor specifications are provided in Appendix B.

### 2.2 Models

We analyze a series of representative open-checkpoint models spanning diverse architectures (autoregressive vs. diffusion-based) and training paradigms (web-data co-training, world modeling, reinforcement learning, etc): (1) `OpenVLA` (Kim et al., 2024) and its variants (2) `OpenVLA-OFT` (Kim et al., 2025), (3) `OpenVLA-OFT_w` (third-view-only version), (4) `OpenVLA-OFT_m` (mix-sft version, trained on all 4 suites), (5) $\pi_0$ (Black et al.), (6) $\pi_0$-`fast` (Pertsch et al., 2025), (7) `Nora` (Hung et al., 2025), (8) `WorldVLA` (Cen et al., 2025), (9) `UniVLA` (Bu et al., 2025) and (10) `RIPT-VLA` (Brohan et al., 2022). Please refer to Appendix C for further details.

### 2.3 Results

We present the main experimental results in Table 1 and Figure 1, which collectively reveal a significant fragility in the generalization capabilities of current VLAs. As shown, even minor perturbations can lead to drastic performance degradation. Below we analyze the specific robustness patterns across perturbation dimensions, models, and tasks.

**Finding 1: Significant Overall Fragility to Perturbations**

Across all perturbation factors, current VLAs exhibit brittle generalization. Performance degrades significantly under various input perturbations, particularly with changes in camera viewpoint and robot initial state.

**Finding 2: Robustness varies considerably by perturbation type.**

Models are most vulnerable to changes in camera viewpoint and robot initial state, which require a high-level understanding of spatial geometry and proprioception. In contrast, they show relative resilience to lighting and background variations, which constitute more superficial, low-level visual changes.

Table 1: Model performance under different perturbations. For each model, the first row reports the task success rate (%) under each perturbation dimension, with the "Original" column indicating the performance on unperturbed inputs. The second row (denoted by ↓) shows the corresponding absolute performance drop. The results highlight significant variations in robustness across models and perturbation types.

| | Original | Camera | Robot | Language | Light | Background | Noise | Layout |
|---|---|---|---|---|---|---|---|---|
| OpenVLA | 76.5 | 1.1 | 4.1 | 26.8 | 4.4 | 25.3 | 19.3 | 31.6 |
| | | ↓75.4 | ↓72.4 | ↓49.7 | ↓72.1 | ↓51.2 | ↓57.2 | ↓44.9 |
| OpenVLA-OFT | 97.1 | 59.7 | 37.2 | 81.5 | 85.8 | 92.4 | 76.7 | 77.1 |
| | | ↓37.4 | ↓59.9 | ↓15.6 | ↓11.3 | ↓4.7 | ↓20.4 | ↓20.0 |
| OpenVLA-OFT_w | 95.3 | 16.8 | 43.7 | 73.2 | 68.2 | 92.5 | 51.4 | 72.3 |
| | | ↓78.5 | ↓51.6 | ↓22.1 | ↓27.1 | ↓2.8 | ↓43.9 | ↓23.0 |
| OpenVLA-OFT_m | 97.6 | 57.9 | 30.6 | 83.6 | 91.6 | 83.6 | 76.3 | 73.2 |
| | | ↓39.7 | ↓67.0 | ↓14.0 | ↓6.0 | ↓14.0 | ↓21.3 | ↓24.4 |
| $\pi_0$ | 94.2 | 15.8 | 6.6 | 61.0 | 79.6 | 78.5 | 79.4 | 70.4 |
| | | ↓78.4 | ↓87.6 | ↓33.2 | ↓14.6 | ↓15.7 | ↓14.8 | ↓23.8 |
| $\pi_0$-fast | 85.5 | 66.4 | 24.8 | 63.3 | 73.0 | 67.7 | 75.8 | 70.3 |
| | | ↓19.1 | ↓60.7 | ↓22.2 | ↓12.5 | ↓17.8 | ↓9.7 | ↓15.2 |
| Nora | 87.9 | 4.0 | 41.1 | 67.0 | 31.0 | 50.5 | 17.6 | 63.9 |
| | | ↓83.9 | ↓46.8 | ↓20.9 | ↓56.9 | ↓37.4 | ↓70.3 | ↓24.0 |
| WorldVLA | 79.1 | 0.3 | 30.2 | 44.2 | 29.4 | 14.5 | 12.2 | 39.4 |
| | | ↓78.8 | ↓48.9 | ↓34.9 | ↓49.7 | ↓64.6 | ↓66.9 | ↓39.7 |
| UniVLA | 95.2 | 4.3 | 50.3 | 71.8 | 59.1 | 80.0 | 25.3 | 34.3 |
| | | ↓90.9 | ↓44.9 | ↓23.4 | ↓36.1 | ↓15.2 | ↓69.9 | ↓60.9 |
| RIPT-VLA | 97.5 | 58.3 | 36.7 | 80.1 | 87.9 | 90.4 | 73.8 | 76.5 |
| | | ↓39.2 | ↓60.8 | ↓17.4 | ↓9.6 | ↓7.1 | ↓23.7 | ↓21.0 |

**Finding 3: Minor Impact of Language Perturbation.**

Contrary to expectations, language perturbations result in the second smallest average performance drop (-25.3) across most models. This apparent robustness is counter-intuitive and merits deeper investigation. As we explore in Section 4, this phenomenon is unlikely to stem from superior linguistic generalization. A more plausible hypothesis, which we have proven empirically, is that models may be relying less on the language instruction than anticipated, potentially leveraging task cues from the visual context.

**Finding 4: Model robustness is dictated by architecture and training paradigm.**

Specifically, models incorporating a first-person wrist camera (e.g., `OpenVLA-OFT`) demonstrate superior generalization, especially to camera viewpoint changes, compared to those reliant solely on a third-person view (e.g., `OpenVLA-OFT_w`). Furthermore, training strategies that emphasize diversity and co-training (e.g., $\pi_0$, $\pi_0$-fast ) consistently yield more robust models across multiple perturbation types, highlighting the importance of exposure to varied data distributions.

# 3 DO CONTEMPORARY VLA MODELS TRULY PAY ATTENTION TO VISUAL INPUTS?

While the overall trends reveal substantial fragility, we observe two particularly interesting patterns in the data: (1) models exhibit surprising resilience to background changes, and (2) several models show limited sensitivity to light variations. These observations raise important questions about what representations the models are actually learning. Do they genuinely understand task-relevant object semantics, or are they relying on superficial visual cues? To answer these questions, we conduct finer-grained analyses of object layout and illumination robustness.

**Do Models Genuinely Attend to Task-Relevant Objects?** We are pleasantly surprised to observe that the models are relatively insensitive to changes in the *Background* setting. To further investigate whether the models truly focus on the core interactive objects and genuinely understand the high-level semantics and spatial information relevant to the task, we decomposed the *Object Layout* perturbation into two subcategories: (1) *adding confounding objects*, and (2) *changing the placement and pose of the target objects*. We then evaluated all models under these conditions, and the results are shown in Figure 1. It can be seen that for $\pi_0$, $\pi_0$-Fast, RIPT-VLA, UniVLA, and World-

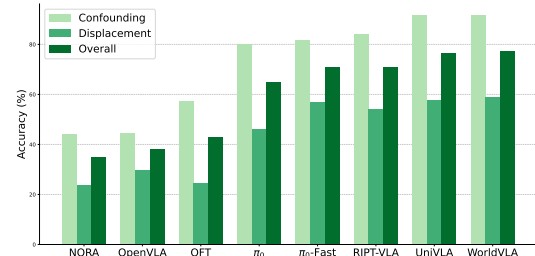

Figure 1: Robustness to object layout perturbations. Comparison of different models under confounding and displacement perturbations, as well as their overall robustness.

VLA, the success rate decreases only marginally when confounding objects are added, indicating that these models, through training, indeed manage to focus their attention on the target objects. However, when the target objects' placement is altered, the performance of the models drops significantly, suggesting that the current models may have merely learned the positional information of the target objects rather than truly capturing the high-level task-relevant semantics.

**How Do Models Maintain Performance Under Illumination Changes?** We observe that for several models, the performance drop under light perturbations is limited to around 10 points, suggesting a surprising insensitivity to illumination changes. To investigate this phenomenon, we design an extreme ablation test: (i) *all-black*, where all camera inputs are replaced with black frames, and (ii) *3rd-black*, where only the third-person view is masked while the wrist camera is preserved. In the *all-black* condition, performance collapses to nearly zero across models, confirming a strong reliance on visual input. In contrast, under the *3rd-black* setting, the same models still achieve accuracies of 43.6, 43.0, and 67.3, respectively, demonstrating that the wrist view alone provides critical and stable close-range geometric and contact cues. This explains why standard light perturbations cause only minor degradation: illumination changes primarily affect the third-person view and global appear-

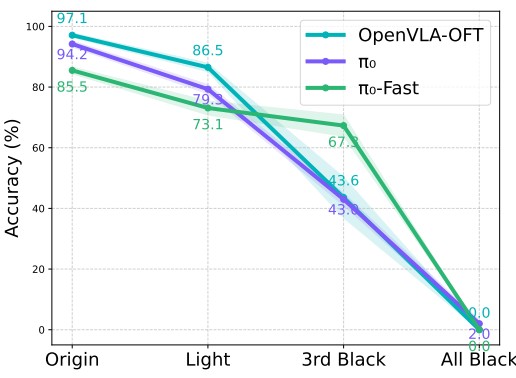

Figure 2: Illumination robustness and extreme ablation tests. The term `Light` denotes the condition with light perturbation applied. `3rd Black` and `All Black` represent conditions where only the third-view image is masked and where images from both views are masked, respectively.

ance, whereas the wrist view remains relatively stable. Consistently, models such as OpenVLA, Nora, and WorldVLA—which depend exclusively on third-person observations—suffer severe drops under light perturbations (often exceeding 60 points).

Based on our deeper investigation into object layout and illumination robustness, we can conclude the following:

**Finding 5: Current VLAs exhibit positional bias rather than genuine semantic understanding of objects.** While models demonstrate an ability to ignore distracting objects, they fail to generalize when target objects are displaced, indicating that they rely on memorized positional cues rather than learning invariant object semantics.

**Finding 6: Wrist cameras provide critical robustness to illumination changes.** The relative stability of performance under light perturbations is largely attributable to the wrist camera's close-range perspective, which provides illumination-invariant geometric cues. Models lacking wrist-camera inputs show significantly greater vulnerability to lighting variations.

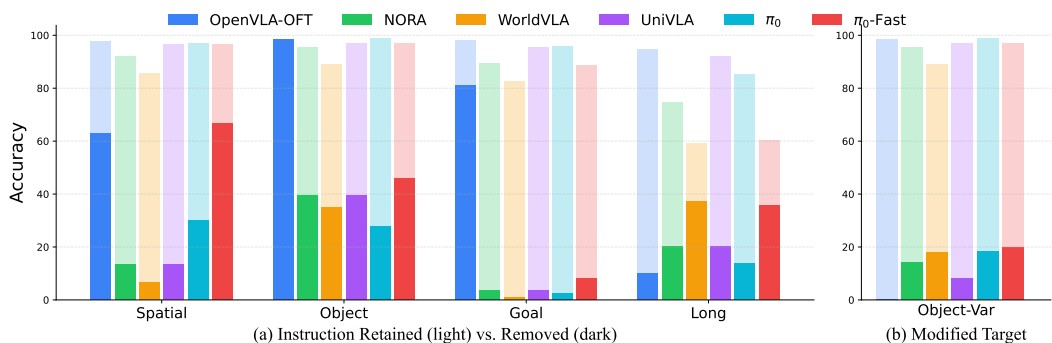

Figure 3: Accuracy of different models on instruction removed (a) and target modified (b) tasks. Light bars: original success rate with language instruction; (a) dark bars: success rate after removing the instruction; (b) Dark bars: success rate under altered task goal and instruction (task substitution).

# 4    DO CONTEMPORARY VLA MODELS TRULY FOLLOW LANGUAGE INSTRUCTIONS?

In the experiments presented in Section 2, we observed an intriguing phenomenon: when introducing language perturbations, the overall performance of the OpenVLA-OFT model was barely affected and remained close to the baseline level. To further investigate the potential underlying reasons, we propose following three hypotheses:

*(1) The model may possess strong generalization capabilities in the language domain*, allowing it to remain robust even when instructions are perturbed.

*(2) The model may extract limited keywords from the input instruction for matching and decision-making*, rather than genuinely understanding the full semantic structure. However, this is unlikely because our perturbations include a *commonsense* subclass that performs keyword commonsense rewrite, yet the performance drop remains nearly negligible.

*(3) The model may not fully utilize the language modality, instead relying primarily on visual or other non-linguistic signals to complete tasks.* In such a scenario, language inputs would be functionally redundant, and even significant perturbations would have minimal impact.

To verify which of the above hypotheses is more plausible, we conducted additional analysis experiments.

## 4.1    WHAT IF WE REMOVE LANGUAGE, DOES PERFORMANCE DROP?

We introduced a **blank instruction** experiment. In this setting, the language input provided to the model was entirely replaced with an empty value, i.e., no linguistic information was supplied during inference. This approach directly tests whether the absence of language leads to a substantial performance degradation. We conducted experiments on all four suites of LIBERO, and the results are shown in Figure 3(a).

Surprisingly, even without any valid language input, the performance of OpenVLA-OFT on the *object* suite remained largely unchanged, with significant degradation observed only on the *long* suite. We attribute this to the greater reliance on instruction guidance in long-horizon tasks, which forces the model to attend to the language modality. This finding is highly revealing: although the model is nominally designed as a Vision-Language-Action (VLA) framework, in practice it degenerates into a form that disregards language, behaving more like a Vision-Action (VA) model.

## 4.2    WHAT IF WE REPLACE GOALS WITH OOD OBJECTS, DO MODELS FAIL?

We further designed a **goal replacement** task to directly examine whether models genuinely possess language instruction-following ability. Specifically, for the *layout* suite, where the issue appeared most pronounced, we replaced the target object in the instruction and the task goal with alternatives within the same scene. For instance, the original task instruction *pick up the alphabet soup*

was replaced with *pick up the tomato sauce*, and similarly, a series of new goal instructions were constructed. As shown in Figure 3(b), the experimental results revealed two key findings:

**Finding 7: VLA models do not possess strong cross-object instruction-following generalization**. In tasks with replaced targets, the model's success rate dropped nearly to zero, with the degradation particularly severe for OpenVLA-OFT. The apparent "robustness" observed in prior language perturbation experiments did not stem from a deep modeling of language but rather from ignoring linguistic inputs altogether, leading to a superficially stable performance under perturbations.

**Finding 8: VLA models appear to rely more on fixed vision–action mappings than on fully exploiting language signals in task decision-making**. By analyzing rollout cases, we observed that even when the target in the instruction was explicitly changed, the model still tended to execute the original target action rather than adjust its behavior according to the new instruction. More details can be found in Appendix F.

## 5 DOES THERE EXIST COMPOSITIONAL GENERALIZATION GAP ACROSS MULTI-DIMENSIONAL PERTURBATIONS?

Generalization results under single-dimension perturbations demonstrate the model's robustness against isolated factors. However, these dimensions may not be independent, and different types of perturbations are likely to exhibit complex dependencies. In this study, we refer to such performance as *compositional generalization*. To ensure scientific rigor, we define the problem from a statistical perspective as follows.

### 5.1 STATISTICAL DEFINITION OF THE COMPOSITIONAL GENERALIZATION GAP

We define the random variables $D_i$ as

$$D_i = \begin{cases} 1, & \text{if the } i\text{-th type of perturbation is applied,} \\ 0, & \text{otherwise,} \end{cases} \tag{1}$$

and similarly for $D_j$. For a single trial, we define the success indicator variable

$$Y = \begin{cases} 1, & \text{if the task is successfully executed,} \\ 0, & \text{otherwise.} \end{cases} \tag{2}$$

The success rate can be defined in terms of conditional probability as

$$s(D_i = d_i, D_j = d_j) = P(Y = 1 \mid D_i = d_i, D_j = d_j), \quad d_i, d_j \in \{0, 1\}. \tag{3}$$

We further estimate the joint probability between $D_i$ and $D_j$ conditioned on $Y = 1$,

$$p(D_i = d_i, D_j = d_j \mid Y = 1) = \frac{s(D_i = d_i, D_j = d_j)}{\sum_{a,b \in \{0,1\}} s(D_i = a, D_j = b)} \tag{4}$$

which represents the probability that the combination $D_i = d_i$ and $D_j = d_j$ occurs among all successful cases. Similarly, the marginal probabilities are

$$p(D_i = 1 \mid Y = 1) = p(D_i = 1, D_j = 0 \mid Y = 1) + p(D_i = 1, D_j = 1 \mid Y = 1)$$
$$= \frac{s(D_i = 1, D_j = 0) + s(D_i = 1, D_j = 1)}{\sum_{a,b \in \{0,1\}} s(D_i = a, D_j = b)}, \tag{5}$$
$$p(D_j = 1 \mid Y = 1) = p(D_i = 0, D_j = 1 \mid Y = 1) + p(D_i = 1, D_j = 1 \mid Y = 1)$$
$$= \frac{s(D_i = 0, D_j = 1) + s(D_i = 1, D_j = 1)}{\sum_{a,b \in \{0,1\}} s(D_i = a, D_j = b)}. \tag{6}$$

Intuitively, $p(D_i = 1 \mid Y = 1)$ reflects the probability that the $i$-th perturbation occurs among all successful cases. It measures the contribution of the $i$-th perturbation to the overall successful outcomes. A high value indicates that the perturbation frequently co-occurs with successful trials,

suggesting the model is robust to this perturbation, while a low value indicates sensitivity to this perturbation. Similarly,

$$p(D_i = 1, D_j = 1 \mid Y = 1) = \frac{s(D_i = 1, D_j = 1)}{\sum_{a,b \in \{0,1\}} s(D_i = a, D_j = b)} \quad (7)$$

represents the proportion of successful cases under the "double perturbation" scenario. A high probability suggests the model maintains performance under joint perturbations, whereas a low probability indicates that the combination severely affects success.

In this study, we focus on the ***Compositionality Gap*** which is also the covariance between variable $D_i$ and $D_j$ given that $Y = 1$:

$$\begin{aligned}
\Delta_{ij} &\triangleq \mathrm{Cov}(D_i, D_j \mid Y = 1) \\
&= \mathbb{E}[D_i D_j \mid Y = 1] - \mathbb{E}[D_i \mid Y = 1]\,\mathbb{E}[D_j \mid Y = 1] \\
&= p(D_i = 1, D_j = 1 \mid Y = 1) - p(D_i = 1 \mid Y = 1)\,p(D_j = 1 \mid Y = 1). \quad (8)
\end{aligned}$$

The sign of $\Delta_{ij}$ correctly reflects the correlation of the contributions of the two perturbations to successful outcomes. Specifically: $\Delta_{ij} > 0$ indicates positive correlation, meaning the model can jointly handle both perturbations. $\Delta_{ij} < 0$ indicates negative interaction, meaning that the combination introduces additional difficulty beyond independent effects. $\Delta_{ij} = 0$ indicates no interaction, satisfying the independence assumption.

## 5.2 EXPERIMENTAL SETUP AND RESULTS ANALYSIS

We perform 2000 independent repeated experiments to ensure high statistical significance. As noted in the previous section, the performance of the VLA model on LLM-Based Language Rewrites is somewhat limited by the model's language-following ability, and its scores may be somewhat "deceptive". Therefore, when analyzing compositional generalization, we select single-dimension perturbations objects spanning, environment sampling, Illumination Variations, camera-sphere shifts, Robot Initialization perturbations, sensor noise and use the OpenVLA-OFT model for testing.

In the experiments, we perform independent tests for each type of single-dimension perturbation and pairwise perturbations, recording the success rate over 2000 repeated trials, which can be found in Appendix G.

The final experimental results are presented in a heatmap shown in Figure 4. The values in the upper-triangular matrix $A_{ij}$ ($1 \le i < j \le 6$) are the product of the conditional probabilities of two single-dimension perturbations. The values in the lower-triangular matrix $A_{ij}$ ($1 \le j < i \le 6$) represent the actual probabilities when applying joint perturbations. Additionally, we calculate the compositional generalization gap

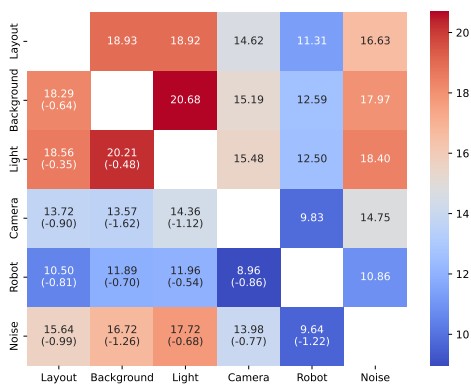

Figure 4: Heatmap of conditional probabilities under pairwise perturbations. Upper triangular entries represent independence-based products of single-dimension probabilities, while lower triangular entries show actual joint outcomes.

$$\Delta_{ij} = A_{ij} - A_{ji} \quad (1 \le j < i \le 6)$$

and verify the statistical significance of the results using a chi-squared test, as shown in Appendix G.

**Finding 9: Generalization is intrinsically non-decomposable**. The consistent negative compositionality gap reflects interaction effects among perturbations, where co-occurring shifts act as coupled noise sources in feature space and expose entanglement in the learned representations. The findings indicate that current models lack mechanisms to capture higher-order dependencies, leading to pronounced robustness degradation under complex perturbation combinations.

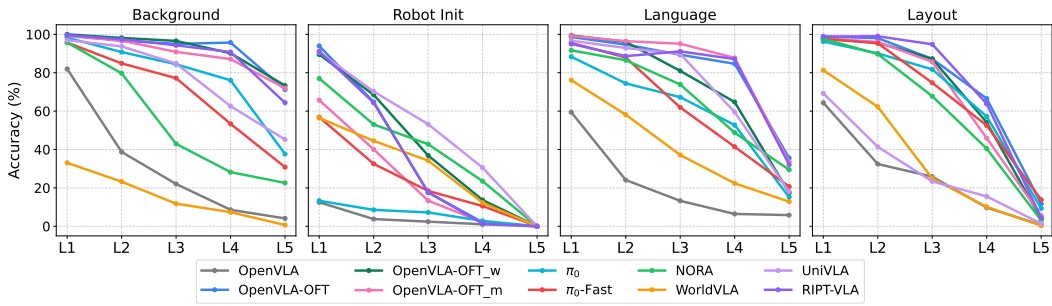

Figure 5: Model performance trends across perturbation difficulty levels. The line plots show the success rate of each model as the intensity of four different perturbation dimensions increases.

# 6 BENCHMARK

## 6.1 BENCHMARK CONSTRUCTION

Building on the analysis in Section 2, we introduce **LIBERO-Pro**, a benchmark designed to rigorously evaluate generalization capabilities along the key dimensions identified in our study. Our construction process consists of two main steps: (1) systematically expanding and enriching the original LIBERO benchmark by applying seven distinct perturbation factors, followed by filtering and balancing task categories based on the findings from Section 2; and (2) evaluating the resulting tasks using four representative models, then stratifying them into five difficulty levels (Level-1–Level-5) according to the accuracy distribution observed across these models. Figure 5 presents the corresponding accuracy of each model across the five difficulty levels under four representative perturbation factors.

The resulting benchmark comprises 10,030 tasks spanning seven perturbation factors with twenty-one low-level components, as illustrated in Figure 6. Detailed generation specifications and level-wise statistics are provided in Appendix E.

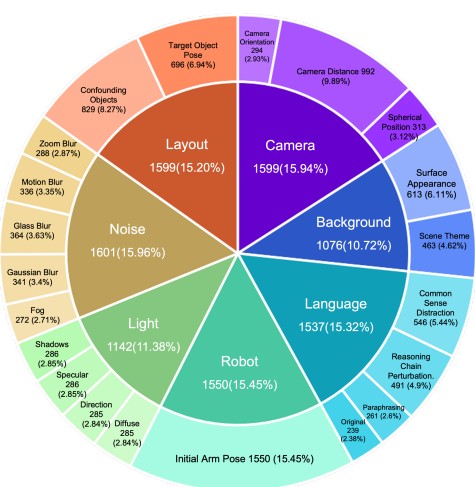

Figure 6: Architecture of the LIBERO-Pro benchmark, comprising 10,030 tasks organized across seven perturbation factors and twenty-one underlying components.

## 6.2 DOES TRAINING ON GENERALIZED SETS IMPROVE GENERALIZATION?

Leveraging our highly automated generalization pipeline, we constructed an extensive training dataset comprising over 20,000 successful trajectories. This dataset was constructed through a substantial expansion of the original LIBERO benchmark, greatly increasing the number of trajectories and scene diversity, enabling a systematic evaluation of how generalization-oriented training affects model performance. Further details on dataset construction are available in Appendix E.

Using this dataset, we conducted mixed fine-tuning starting from the official OpenVLA-OFT weights. The corresponding results on the LIBERO-pro benchmark are presented in Table 2.

As shown in Table 2, our method achieves the highest overall success rate (79.6%), outperforming all baseline models across nearly all perturbation types. Most notably, it exhibits a dramatic improvement in camera view robustness (92.8%), surpassing the next best model by 37.2 percentage points. Significant gains are also observed under noise (89.3%) and layout (77.6%) perturbations. These results demonstrate that training with our generalized dataset substantially enhances model robustness to a wide range of unseen environmental variations.

Table 2: Robustness evaluation across perturbation dimensions, with **bold** values denoting the highest scores. The bottom row (+ PT) shows the performance of our post-training method, with absolute improvements over the baseline (OpenVLA-OFT_m) indicated by upward arrows.

| | Camera | Robot | Language | Light | Background | Noise | Layout | Total |
|---|---|---|---|---|---|---|---|---|
| OpenVLA | 0.8 | 3.5 | 23.0 | 8.1 | 50.4 | 15.2 | 28.5 | 17.3 |
| OpenVLA-OFT | 56.4 | 31.9 | 79.5 | 88.7 | 97.3 | 75.8 | 74.2 | 70.0 |
| OpenVLA-OFT_w | 10.4 | 38.7 | 70.5 | 76.8 | 99.2 | 49.9 | 69.9 | 56.4 |
| NORA | 2.2 | 37.0 | 65.1 | 45.7 | 65.5 | 12.8 | 62.1 | 39.8 |
| WorldVLA | 0.1 | 27.9 | 41.6 | 43.7 | 19.8 | 10.9 | 38.0 | 25.3 |
| UniVLA | 1.8 | 46.2 | 69.6 | 69.0 | 90.7 | 21.2 | 31.9 | 43.9 |
| $\pi_0$ | 13.8 | 6.0 | 58.8 | 85.0 | 90.7 | 79.0 | 68.9 | 54.6 |
| $\pi_0$-Fast | 65.1 | 21.6 | 61.0 | 73.2 | 97.7 | 74.4 | 68.8 | 64.2 |
| RIPT-VLA | 55.2 | 31.2 | 77.6 | 88.4 | **100.0** | 73.5 | 74.2 | 69.3 |
| Openvla-OFT_m | 55.6 | 21.7 | 81.0 | 92.7 | 92.3 | 78.6 | 68.7 | 68.1 |
| Ours | **92.8** | **30.3** | **85.8** | **94.9** | 93.9 | **89.3** | **77.6** | **79.6** |
| | ↑37.2 | ↑8.6 | ↑4.8 | ↑2.2 | ↑1.6 | ↑10.7 | ↑8.9 | ↑11.5 |

# 7 RELATED WORK

## 7.1 VISION-LANGUAGE-ACTION MODELS

The paradigm of foundation models has recently extended from language and vision into robotics, motivating unified architectures that couple perception, language understanding, and control in an end-to-end foundation model . Autoregressive approaches [Brohan et al. (2022), Kim et al. (2024), Pertsch et al. (2025), Li et al. (2025), Wen et al. (2025a), Li et al. (2024b)] discretize robot actions into tokens and train end-to-end policies on large-scale demonstrations, while diffusion-based models [Black et al., Bjorck et al. (2025), Li et al. (2024a), Wen et al. (2025b)] generate continuous trajectories via generative diffusion experts. More recently, reinforcement learning methods [Tan et al. (2025), Liu et al. (2025), Lu et al. (2025), Guo et al. (2025)] move beyond supervised fine-tuning, emphasizing robustness and downstream adaptability through reinforcement learning objectives. Although these models can exhibit "zero-shot" competence when evaluation closely resembles training conditions—e.g., similar object layouts, robot initial poses, or camera viewpoints—such results primarily reflect interpolation rather than genuine robustness. Existing benchmarks thus provide limited insight into how these models generalize under controlled distribution shifts, leaving the lack of a systematic and fine-grained robustness analysis as a central open challenge.

## 7.2 GENERALIZATION ROBOTIC MANIPULATION EVALUATIONS

How to efficiently and systematically evaluate the generalization ability of robotic manipulation models remains a central question in the field. Early efforts (Liu et al. (2023), James et al. (2020), Mu et al. (2021)) provided reproducible environments and benchmarks. More recent work (Zhou et al. (2025), Garcia et al. (2025)) introduced perturbations along the object dimension, such as modifying target objects, altering positions, or adding confounding items. Other benchmarks (Liu et al. (2025), Fang et al. (2025)) extended evaluation to additional factors, including task language instructions, robot initial states, and environmental backgrounds. However, these approaches rely heavily on manually designed tasks and perturbations, which limits scalability and constrains the total number of tasks to only a few dozen, resulting in insufficient systematic coverage. More automated frameworks (Wang et al. (2025), Pumacay et al. (2024)) significantly increase task counts and incorporate broader dimensions such as illumination and camera viewpoints. Nevertheless, they lack fine-grained analysis within each dimension, and thus the insights they provide remain limited.

# 8 CONCLUSION

This work systematically analyzes modern VLA models, exposing a significant generalization problem in contrast to their alomose saturated performance on benchmarks such as LIBERO. Our findings reveal that most of the contemporary VLA models remain brittle, showing particular vulnerability to camera and robot state changes, almost all models ignore the language instructions, and some of the models execute with a bare memorization of the trajectory instead of relying on visual feedbacks. We also identify positional bias and negative combinatorial generalization gaps under combined perturbations. We urge the community to prioritize the true diversity of embodied tasks in evaluation and develop architectures capable of robust generalization beyond limited benchmark environments.

## ETHICS STATEMENT

This work adheres to the ICLR Code of Ethics. In this study, no human subjects or animal experimentation was involved. All datasets and benchmarks used, including LIBERO(Liu et al., 2023), were sourced in compliance with relevant usage guidelines, ensuring no violation of privacy. We have taken care to avoid any biases or discriminatory outcomes in our research process. No personally identifiable information was used, and no experiments were conducted that could raise privacy or security concerns. We are committed to maintaining transparency and integrity throughout the research process.

## REPRODUCIBILITY STATEMENT

We have made every effort to ensure that the results presented in this paper are reproducible. All code and datasets have been made publicly available in an anonymous repository to facilitate replication and verification. The experimental setup, including training steps, model configurations, and hardware details, is described in detail in the paper. We have also provided a full description of model evaluation, perturbation injection and dataset construction, to assist others in reproducing our experiments. For implementation detail and code, please refer to our anonymous GitHub repository: `https://anonymous.4open.science/r/LIBERO-pro-522F`.

Additionally, Vision-Language-Action (VLA) model training datasets, such as LIBERO(Liu et al., 2023), are publicly available, ensuring consistent and reproducible evaluation results.

We believe these measures will enable other researchers to reproduce our work and further advance the field.

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

## A    LLM USAGE

Large Language Models (LLMs) were used to aid in the writing and polishing of the manuscript. Specifically, we used an LLM to assist in refining the language, improving readability, and ensuring clarity in various sections of the paper. The model helped with tasks such as sentence rephrasing, grammar checking, and enhancing the overall flow of the text.

It is important to note that the LLM was not involved in the ideation, research methodology, or experimental design. All research concepts, ideas, and analyses were developed and conducted by the authors. The contributions of the LLM were solely focused on improving the linguistic quality of the paper, with no involvement in the scientific content or data analysis.

The authors take full responsibility for the content of the manuscript, including any text generated or polished by the LLM. We have ensured that the LLM-generated text adheres to ethical guidelines and does not contribute to plagiarism or scientific misconduct.

## B    PERTURBATION DIMENSIONS

We conducted a comprehensive review of existing studies aimed at evaluating the generalization performance of VLA models, particularly those introducing new test suites such as COLOSSEUM (Pumacay et al., 2024), RL4VLA (Liu et al., 2025), AGNOSTOS (Zhou et al., 2025), *etc.* The comparison of these methods is summarized in Table 3. Based on a systematic analysis of their task paradigms, environment construction, data collection pipelines, and evaluation dimension designs, this study ultimately identified seven core dimensions of perturbation: objects layout, environment background sampling, light variations, camera-view shifts, robotarm initialization perturbations, LLM-based language rewrites, and image noise, with the goal of testing model robustness and generalization ability across all modalities of input (vision, state, language). Each dimension contains multiple quantifiable sub-dimensions defined to enable fine-grained evaluation of model performance.

The perturbed examples are shown in Figures 11–16.

Table 3: Comparison of Different Evaluation Methods for VLA Models

| Method | Automation | Simulator | Fine-grained | Perturbation Dimensions | | | | | | |
|---|---|---|---|---|---|---|---|---|---|---|
| | | | | Obj | Background | Light | Camera | Robot | Language | Noise |
| AGNOSTOS | × | RLBench | × | ✓ | × | × | × | × | × | × |
| VLATest | ✓ | ManiSkill | × | ✓ | × | ✓ | ✓ | × | ✓ | × |
| RL4VLA | × | ManiSkill | × | ✓ | ✓ | × | × | ✓ | ✓ | × |
| INT-ACT | × | ManiSkill | × | ✓ | × | × | × | × | ✓ | × |
| Gembench | × | RLBench | × | ✓ | × | × | × | × | × | × |
| COLOSSEUM | ✓ | RLBench | × | ✓ | ✓ | ✓ | ✓ | × | × | × |
| LIBERO-pro (Ours) | ✓ | LIBERO | ✓ | ✓ | ✓ | ✓ | ✓ | ✓ | ✓ | ✓ |

### B.1    OBJECTS LAYOUT

This dimension is designed to test model robustness against object-level disturbances. It is further divided into two sub-dimensions:

- **O1: Confounding Objects.** Randomly add $n$ additional unseen objects into the task scene. The object categories are drawn from a predefined set of 416 distractor objects. This perturbation is implemented by modifying the task description files (BDDL). In the benchmark, related perturbations are stored in BDDL files with an *add* suffix.

- **O2: Target Object Pose.** Apply random perturbations to the target object's initial position $(x, y, z)$ and orientation (pitch, yaw, roll). This perturbation does not alter the target object itself and ensures that essential semantic relations to other objects remain unchanged (e.g., in the task pick_up_the_black_bowl_next_to_the_cookie_box_and_place_it_on_the_plate, the relation to the cookie box determines the target object, and our modifications do not alter this constraint).

## B.2 BACKGROUND TEXTURES

This dimension evaluates the model's ability to generalize to different background textures of the scene. It contains two sub-dimensions:

- **B1: Scene Theme.** Change the scene texture of the environment (e.g., from painted wall to brick wall). The new textures are sourced from a curated collection of 950 textures. This perturbation is implemented by modifying the scene XML definition files and registering new scene classes.
- **B2: Surface Appearance.** Randomly alter the texture of the working surface (e.g., tabletop or floor).

## B.3 LIGHT CONDITIONS

This dimension evaluates the model's visual understanding under different lighting conditions. It includes four sub-dimensions, all implemented by modifying scene XML definition files:

- **L1: Diffuse.** The diffuse color, which defines the light color uniformly reflected by object surfaces (adjusted via RGB channels; e.g., 1 0 0 indicates red diffuse light, making objects appear reddish).
- **L2: Direction.** Change the direction of the parallel light source, which significantly affects color rendering and shading.
- **L3: Specular.** The intensity of the specular highlight on object surfaces (e.g., the bright spot reflected on metals). Larger values yield more distinct highlights, strongly influencing scene style.
- **L4: Shadows.** Boolean variable (true/false) indicating whether shadows of the robot arm and objects are cast in the scene.

## B.4 CAMERA VIEWPOINTS

This dimension tests the model's view-free representation and generalization ability by changing camera viewpoints. All perturbations are implemented by modifying the *Problem* class interface, with parameters derived from task filenames:

- **C1: Camera Distance.** Move the camera along its optical axis, changing the distance to the scene center. Camera distances are valued among $1.01\times$ to $2.00\times$ the original value.
- **C2: Spherical Position.** Perturb camera position on a sphere centered at the scene, altering azimuth ($\Delta\theta$) and elevation ($\Delta\phi$) within $15°$–$75°$ cones.
- **C3: Camera Orientation.** Fix the camera position but perturb its orientation (yaw, pitch, roll), valued within $2°$ to $10°$.

## B.5 ROBOT INITIAL STATES

- **Initial Joint Angle.** Random perturbations are applied to the robot arm's initial joint positions (qpos). Perturbation magnitudes are valued from 0.1 to 0.5. This perturbation is implemented by modifying the *Problem* class interface.

## B.6 LANGUAGE INSTRUCTIONS

This dimension employs large language models (LLMs) to rewrite original task instructions, testing model generalization and reasoning ability in natural language:

- **R1: Distraction.** Task instructions are rewritten into longer and more conversational forms that contain additional but task-irrelevant contextual cues.
- **R2: Common Sense.** Replacing the existing object descriptions with commonsense-based descriptions to test information extraction and filtering.

- **R3: Reasoning Chain.** For multi-step reasoning instructions, perturbations involve altering reasoning complexity.

Table 4: Examples of Language Instruction Rewriting

| Sub-category | Examples |
|---|---|
| Original | push the plate to the front of the stove |
| R1 | before turning on the burner, push the plate to the front of the stove |
| R2 | propel the flat surface used for holding food toward the area designated for cooking heat adjustment |
| R3 | make sure the plate ends up at the front of the stove |

### B.7 SENSOR NOISE

This dimension simulates real-world sensor imperfections to evaluate robustness under degraded input quality:

- **N1: Motion Blur.** Simulates blur caused by relative motion between camera and scene. Higher levels correspond to larger blur kernels, longer trajectories, and more severe blur.

- **N2: Gaussian Blur.** Simulates optical blur caused by defocus. Higher levels correspond to larger kernel size and standard deviation, resulting in smoother images with greater detail loss.

- **N3: Zoom Blur.** Simulates radial blur caused by rapid zoom during exposure. Higher levels increase zoom center and blur intensity, producing strong vignetting.

- **N4: Fog.** Simulates atmospheric interference such as fog or haze. Higher levels increase fog density and brightness, lowering image contrast and saturation.

- **N5: Glass Blur.** Simulates distortions and refractions caused by viewing through textured glass. Higher levels increase distortion amplitude and range, resulting in severe local pixel displacements.

Perturbation parameters are shown in Table 5.

## C MODEL DETAILS

This appendix provides comprehensive descriptions of all models evaluated in our study, covering their architectural designs, training data sources, and key implementation specifications. We aim to offer sufficient transparency such that the reported results can be faithfully reproduced and compared against future work.

### C.1 MODEL OVERVIEW

We evaluate a diverse set of vision-language-action (VLA) models that represent different design choices in terms of architecture and training strategy, enabling us to systematically analyze how different factors contribute to task performance and robustness. For each model, we summarize its backbone, modality encoders, fusion mechanisms, and decision heads.

### C.2 OPENVLA(KIM ET AL., 2024) AND OPENVLA-OFTS(KIM ET AL., 2025)

**Base Architecture.** OpenVLA adopts a modular vision-language architecture built on the Prismatic-7B VLM. The visual encoder is a 600M-parameter dual-backbone composed of SigLIP and DINOv2, whose outputs are concatenated along the channel dimension to enhance spatial reasoning capabilities crucial for robotic control. A lightweight two-layer MLP projector maps the fused visual features into the input space of a Llama2-7B language backbone, which integrates visual and textual inputs through cross-attention. This design enables OpenVLA to leverage both semantic

Table 5: Noise perturbation parameters.

| ID | Noise Type | Key Parameters | Description of L1–L5 |
|---|---|---|---|
| 1 | Motion Blur | Radius $r$, Gaussian kernel $\sigma$, angle $\theta$ | $r$ and $\sigma$ control blur strength (kernel size and spread). From weak blur ($r = 5, \sigma = 2$) to strong blur ($r = 35, \sigma = 20$). $\theta \sim U(-45°, 45°)$ determines blur direction. |
| 2 | Gaussian Blur | Standard deviation $\sigma$ | $\sigma$ controls the amount of smoothing. Small $\sigma$ produces slight blur ($\sigma = 1$), large $\sigma$ produces heavy blur ($\sigma = 10$). |
| 3 | Zoom Blur | Scaling factors $[s_{min}, s_{max}, step]$ | Successive rescaling creates a zoom-like blur. Weak effect at ($[1, 1.11, 0.01]$) and strong effect at ($[1, 1.56, 0.03]$). |
| 4 | Fog | Density $\alpha$, decay rate $\beta$ | $\alpha$ controls fog thickness, $\beta$ controls how quickly fog attenuates. Light fog ($\alpha = 0.5, \beta = 3.0$) $\rightarrow$ Dense fog with slow decay ($\alpha = 5.0, \beta = 1.3$). |
| 5 | Glass Blur | Gaussian blur $\sigma$, pixel displacement $\delta$, iteration count | $\sigma$ defines baseline blur, $\delta$ controls the displacement of pixels, and iterations determine the accumulation of distortions. Light blur with small displacements ($\sigma = 0.5, \delta = 1, iters = 3$) $\rightarrow$ Strong blur with large displacements ($\sigma = 2.5, \delta = 5, iters = 1$). |

Table 6: Model HuggingFace repository addresses.

| ID | Model Name | Checkpoint Address |
|---|---|---|
| 1 | OpenVLA | OpenVLA LIBERO checkpoint |
| 2 | OpenVLA-OFT | OpenVLA-OFT LIBERO checkpoint |
| 3 | OpenVLA-OFT_m | OpenVLA-OFT_m LIBERO checkpoint |
| 4 | NORA | NORA LIBERO checkpoint |
| 5 | WorldVLA | WorldVLA LIBERO checkpoint |
| 6 | UniVLA | UniVLA LIBERO checkpoint |
| 7 | $\pi_0$ | $\pi_0$ LIBERO checkpoint |
| 8 | $\pi_0$-Fast | $\pi_0$-Fast LIBERO checkpoint |
| 9 | RIPT-VLA | RIPT-VLA LIBERO checkpoint |

understanding and spatial grounding for action prediction. To adapt the VLM backbone for robotic control, continuous robot actions are discretized into 256 bins per dimension and represented as tokens within the LLM vocabulary. The 256 least frequently used tokens of the Llama tokenizer are replaced by action tokens, and training proceeds with the standard next-token prediction objective applied to action sequences.

**Training Strategy.** The training pipeline consists of two stages: an initial pre-training followed by supervised fine-tuning. OpenVLA is pre-trained on the Open X-Embodiment (OpenX) dataset, which includes over 970k robot trajectories across multiple embodiments and tasks. The model is trained end-to-end with a cross-entropy loss applied exclusively to the action tokens. Unlike typical VLM practices, the vision encoder is fine-tuned rather than frozen, enabling the model to capture fine-grained spatial details crucial for robotic control.

**Variants.** In addition to the baseline OpenVLA models, we consider the OpenVLA-OFT family of variants:

- **OpenVLA-OFT:** A parallel decoding variant enabling simultaneous prediction of all actions in a single forward pass. It employs continuous action representations through a multi-layer MLP head and is trained with an L1 regression objective, resulting in faster inference

and more precise action generation, and it incorporates Feature-wise Linear Modulation (FiLM) to enhance language grounding.

- **OpenVLA-OFT_w:** A variant of OpenVLA-OFT that removes the first-person wrist camera input and retains only the third-person view. This model is trained from OpenVLA with the official OFT hyperparameters on four LIBERO benchmark suites for 150K steps using 8×A100 GPUs.

- **OpenVLA-OFT_m:** A mixed-training variant that adopts the official mix-SFT weights. Unlike suite-specific training, this model is jointly trained across all four LIBERO suites, enabling it to learn from a broader distribution of tasks and environments.

## C.3 $\pi_0$(BLACK ET AL.) AND $\pi_0$-FAST(PERTSCH ET AL., 2025)

**Base Architecture.** The $\pi_0$ architecture is inspired by the Transfusion framework, which trains a single Transformer with multiple objective functions: a flow-matching loss for continuous output tokens and a cross-entropy loss for discrete tokens. Building upon this, $\pi_0$ implements two sets of transformer weights (one initialized from the VLM and a smaller action expert). The core model comprises a VLM base (PaliGemma) for semantic understanding of multimodal inputs (multiple RGB images, language instructions, and proprioceptive state $q_t$), and action tokens are projected and routed to a smaller action-expert.

**Training Strategy.** The training follows a two-stage paradigm: (i) large-scale, diverse pre-training on a mixture ddataset to learn broad capabilities and recovery behaviors; (ii) post-training on smaller, high-quality curated datasets to induce dexterity and fluent task execution. The pre-training mixture is carefully reweighted to avoid over-representation.

**$\pi_0$-fast: Efficient Action Tokenization.** The $\pi_0$-fast variant introduces the FAST tokenization method to compress action sequences. FAST combines a Discrete Cosine Transform (DCT) for converting temporal action trajectories into a sparse frequency-domain representation, followed by Byte-Pair Encoding (BPE) to losslessly compress the sparse DCT coefficient matrix into dense tokens.

## C.4 NORA(HUNG ET AL., 2025)

**Base Architecture.** NORA is a 3B-parameter general-purpose VLA model optimized for robotic tasks. It adopts the Qwen-2.5-VL-3B multimodal model as its backbone, chosen for its strong visual-semantic understanding capabilities, which enhance visual reasoning and action grounding. The model processes natural language task instructions and single-frame (as per its implementation) visual observations as input. It outputs discrete action sequences by employing the FAST+ tokenizer to discretize continuous action tokens.

**Training Strategy.** NORA is pre-trained on the Open X-Embodiment dataset, which includes trajectories from various robots performing diverse tasks. This phase aims to equip the model with broad robotic capabilities and strong generalization. Training was conducted on 8×H100 GPUs for approximately three weeks (totaling 4000 GPU hours), using the AdamW optimizer with a batch size of 256 over 1.1 million gradient steps. A linear warmup followed by cosine decay was applied to the learning rate.

## C.5 WORLDVLA(CEN ET AL., 2025)

**Base Architecture.** WorldVLA is an autoregressive action-world model that unifies visual-language-action (VLA) modeling and world modeling within a single, integrated framework. The core idea is to jointly learn a policy model for action generation and a world model for future state prediction, allowing the two components to mutually enhance each other. The model is initialized from Chameleon, a unified image understanding and generation model. It employs three tokenizers: a VQ-GAN-based image tokenizer, a BPE-based text tokenizer, and an action tokenizer that discretizes each dimension of the continuous robot action into 256 bins. All modalities (text, image, action) are discretized into tokens and modeled autoregressively within a unified sequence. A key architectural innovation is a customized attention mask for action generation that prevents the

current action from attending to previous actions, thereby mitigating error propagation and enabling more robust parallel action chunk prediction.

**Training Strategy.** The model is trained on a mixture of action-modeling data and world-modeling data. The action-modeling data trains the model to generate action chunks given a language instruction and a history of image observations, using a loss computed only on the action tokens. The world-modeling data trains the model to predict the next image frame given the current image and action, using a loss computed only on the image tokens. This joint training strategy encourages the learning of shared representations: the world model acquires an understanding of environmental physics to aid task-relevant action generation, while the action model enhances visual understanding to support accurate frame prediction. The model is evaluated on the LIBERO benchmark suite, with training leveraging 90% of the successful trajectories for training and 10% for validation.

### C.6 UniVLA (Li et al., 2025)

**Base Architecture.** UniVLA is a universal visual-language-action model that operates in a discrete, task-centric latent action space to achieve cross-embodiment generalization. The architecture is built upon a pre-trained Prismatic-7B VLM, which integrates a fused visual encoder (SigLip and DINOv2), a projection layer, and an LLaMA-2 LLM. A key innovation is the extension of the LLM's vocabulary with special action tokens to represent quantized latent actions. The model takes a visual observation and a language instruction as input and autoregressively predicts a sequence of these latent action tokens. For deployment on specific robots, a lightweight action decoder head is added, which uses multi-head attention pooling to map the predicted latent actions into the robot's executable low-level control space.

**Training Strategy.** The training process involves three stages. First, a latent action model is trained in a self-supervised manner on large-scale video datasets to learn a discrete codebook of task-centric actions. This model uses a DINOv2-based reconstruction objective and conditions on language instructions to disentangle task-relevant dynamics from irrelevant visual changes. Second, the universal policy is pre-trained to predict these latent action tokens from observations and instructions, leveraging the generalizable representations of the pre-trained VLM. This approach compresses the action space dramatically, leading to significantly faster convergence compared to methods operating in raw action spaces. Finally, for downstream adaptation, the entire model is fine-tuned end-to-end with a combined loss for latent action prediction and low-level action regression, often using parameter-efficient methods like LoRA. A history-augmented input scheme, where past latent actions are fed back as context, is employed to enhance performance in long-horizon tasks.

### C.7 RIPT-VLA (Brohan et al., 2022)

**Base Architecture.** The base model for RIPT-VLA is OpenVLA-OFT, a continuous-action VLA model where the action head is typically trained with an L1 regression loss. To make this architecture compatible with reinforcement learning, which requires a probabilistic policy output, RIPT-VLA augments the model with a lightweight auxiliary head that predicts the scale parameter $\sigma_\theta$ for the action distribution. The policy is then treated as a factorized Laplace distribution (for L1 loss) with the original model output as the mean $\mu_\theta$ and the new head's output as the scale. This allows for sampling actions and computing the log-probability $\log \pi_\theta(a_t|a_{<t}, c)$ in closed form, which is essential for policy gradient updates.

**Training Strategy.** RIPT-VLA introduces a third stage of Reinforcement Interactive Post-Training (RIPT) following the standard pre-training and supervised fine-tuning (SFT) stages. The strategy is centered on the Dynamic Sampling Leave-One-Out PPO (LOOP) framework. In the rollout collection phase, for a given context $c_i$, $K$ trajectories are sampled from the current policy. The RLOO advantage estimation method is used to compute advantages from the sparse binary rewards. A key innovation is a dynamic rejection mechanism that filters out context samples where all $K$ rollouts receive identical rewards (all successes or all failures), thus ensuring that the training batch contains meaningful learning signals. During the policy optimization phase, the PPO algorithm is applied to the collected rollouts to maximize the expected task success rate, with the policy update constrained by the probability ratio $r_i = \pi_\theta(a_i|c_i)/\pi_\psi(a_i|c_i)$ to ensure stable training. This iterative process of data collection and optimization allows the model to improve its performance through environment interaction, specifically targeting and overcoming failure modes encountered during deployment.

## D    Perturbations and Benchmark Construction

### D.1    Data Generation and Filtering

We began with the 40 evaluation tasks from LIBERO and generated 500 instances for each of the four generalization sub-tasks (Spatial, Object, Goal, Long) across the seven generalization dimensions, resulting in an initial set of 14,000 candidate tasks. These tasks were evaluated using several widely adopted baseline models to assess performance distributions, as summarized in Section 2.

Tasks that were solved by all models, or by a large majority, were removed to avoid ceiling effects. We further balanced the remaining tasks across augmentation sub-dimensions to prevent bias. The final test-only benchmark consists of 10,030 tasks spanning all seven dimensions.

### D.2    Dataset Composition

Table 7 presents the final distribution of evaluation tasks across generalization dimensions and sub-task categories.

Table 7: Distribution of the evaluation dataset across dimensions and different categories.

|         | Camera | Robot | Language | Light | Background | Noise | Layout | Total |
|---------|--------|-------|----------|-------|------------|-------|--------|-------|
| Spatial | 376    | 350   | 354      | 292   | 258        | 351   | 312    | 2293  |
| Object  | 396    | 398   | 390      | 297   | 248        | 422   | 425    | 2576  |
| Goal    | 408    | 409   | 410      | 279   | 281        | 379   | 403    | 2569  |
| Long    | 419    | 393   | 383      | 274   | 289        | 449   | 385    | 2592  |
| Total   | 1599   | 1550  | 1537     | 1142  | 1076       | 1601  | 1525   | 10030 |

### D.3    Difficulty Assessment

We evaluated the 10,030 tasks using four representative models—Openvla-oft, pi0, pi0-fast, and uni-vla—and stratified task difficulty based on how many of these models succeeded on each instance:

- Level 1 (lv1): solved by all four models;
- Level 2 (lv2): solved by exactly three models;
- Level 3 (lv3): solved by exactly two models;
- Level 4 (lv4): solved by exactly one model;
- Level 5 (lv5): solved by none.

Figure 7 illustrates the proportion of tasks at each difficulty level for every dimension.

### D.4    Model Performance by Difficulty Level

We further analyzed how model accuracy varies with task difficulty. Figure 8 shows the success rates of each model across the five difficulty levels.

## E    Training Dataset Construction

### E.1    Dataset Overview

The generalized training dataset consists of over 20,000 successful trajectories, covering a wide range of task variations and environment configurations. Figure 9 shows the distribution of the 7-dimensional robot actions in the dataset. The plots are arranged from top to bottom and left to right, corresponding to the seven action dimensions, respectively. This visualization demonstrates the diversity and coverage of the actions captured in the generalized dataset.

The dataset includes six types of task variants and environment modifications: objects spanning, environment sampling, light variations, camera-view shifts, LLM-based language rewrites, and sensor noise. Among these, the objects spanning variant contains only compounding objects, which

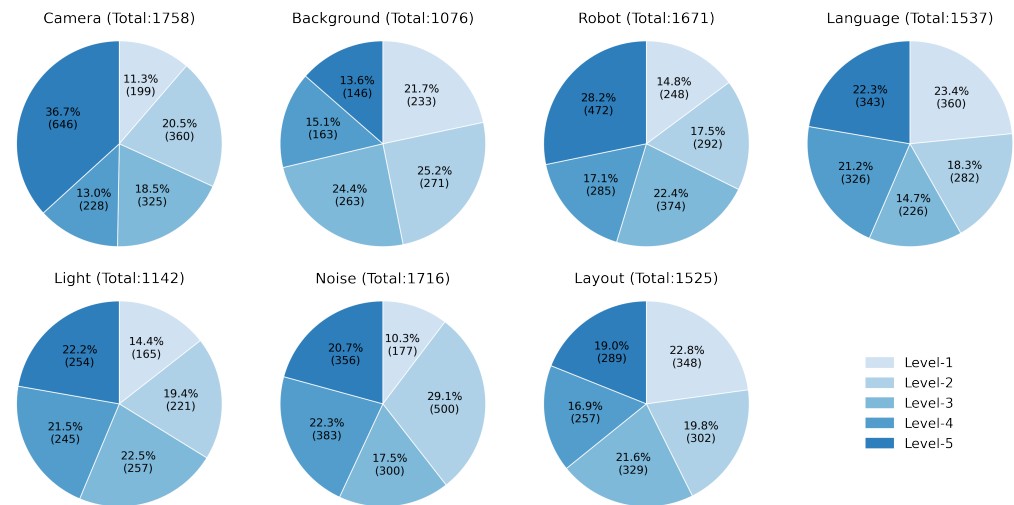

Figure 7: Proportion of tasks per difficulty level across the seven generalization dimensions.

are generated by executing existing trajectories and selecting only the successful ones. Variants involving pose changes were not added due to the limited reliability of automatically generated trajectories.

### E.2 DATA GENERATION PROCESS

The generalized dataset was constructed using the same automated generalization pipeline, with variations in parameters to produce diverse scenarios:

- **Objects Positioning:** For compounding objects, distractor objects and their poses were varied while ensuring no overlap with the test set.
- **Background Environment Sampling:** Additional textures for tables, walls, and floors were automatically sampled to avoid overlap with the test environments.
- **Light Variations:** Different lighting parameters were applied to the scenes.
- **Camera-view Shifts:** Camera angles differed by 5° on the spherical coordinate system compared to the test set.
- **LLM-based Language Rewrites:** New language instructions were generated to provide additional linguistic diversity.
- **Image Noise:** Sensor noise parameters differed from those listed in Table 5.

### E.3 TRAJECTORY COLLECTION

Trajectory collection was performed using the original LIBERO dataset's (state, action) pairs, executed in the newly generated environments. Only successful trajectories were retained, and any actions corresponding to no-ops were filtered out. Specifically, 2,400 trajectories were collected for the compounding object variant, while 4,000 trajectories were collected for each of the other variants, resulting in a total of 22,400 trajectories. After filtering, over 20,000 high-quality trajectories were retained for training.

### E.4 TRAINING CONFIGURATION

Using this dataset, we performed mixed fine-tuning based on the official OpenVLA-OFT weights. The training was conducted on $8 \times$ A100 GPUs with a learning rate of $5 \times 10^{-4}$ for 100,000 steps. The batch size was set to 64 per GPU, resulting in an effective batch size of 512. We employed the AdamW optimizer with weight decay of 0.1 and used a cosine learning rate schedule with warmup. The training results on LIBERO-pro are shown in Table 2.

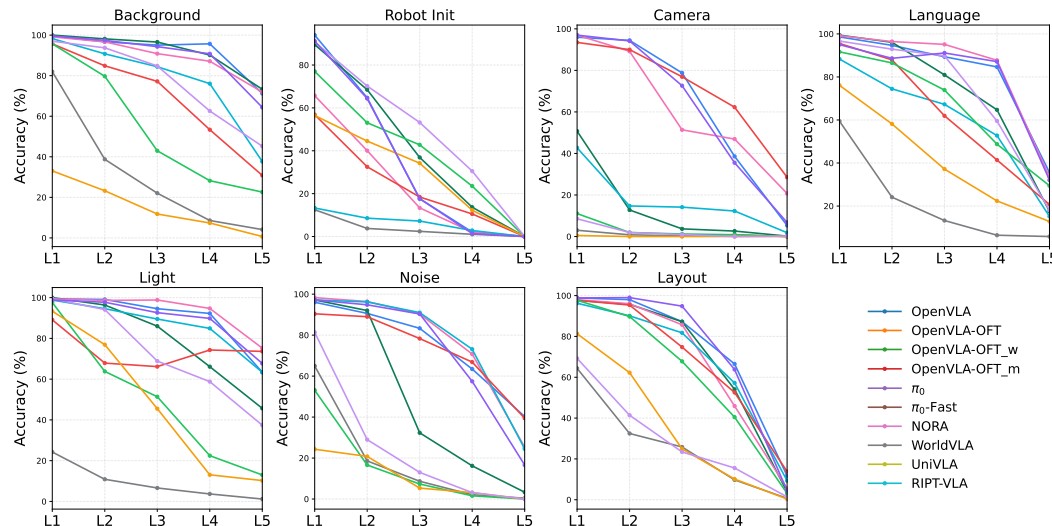

Figure 8: Model performance trends across perturbation difficulty levels. The line plots show the success rate of each model as the intensity of all seven different perturbation dimensions increases.

### E.5 STORAGE FORMAT

All trajectories are stored in the `rlds` format, consistent with standard practices for robotics datasets and ensuring compatibility with existing training pipelines.

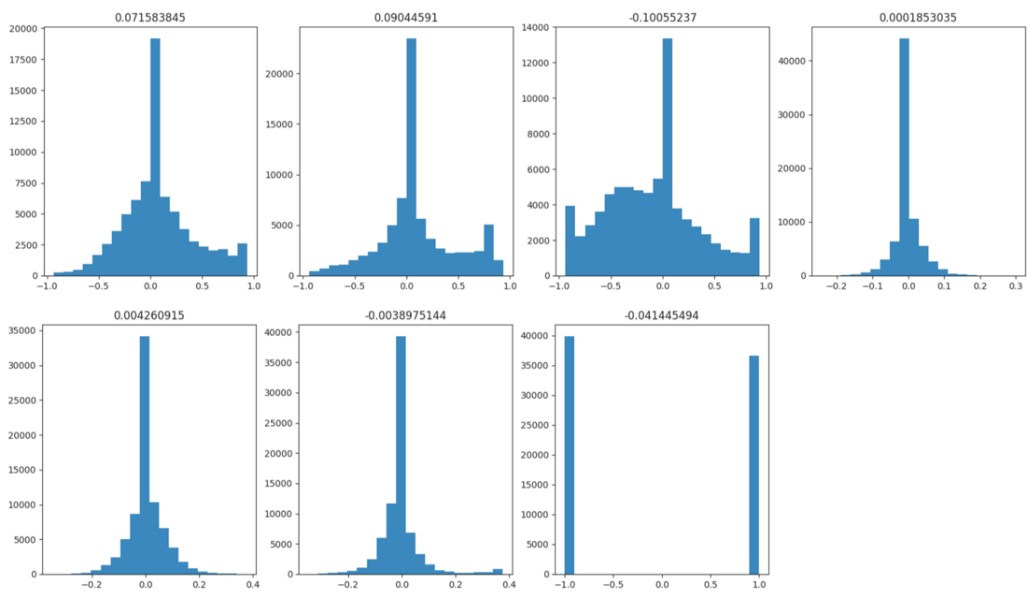

Figure 9: Distribution of the 7-dimensional robot actions in the generalized dataset. Plots are arranged from top to bottom and left to right, corresponding to action dimensions 1–7.

## F GOAL REPLACEMENT ROLLOUT CASES ANALYSIS

To further probe whether Vision-Language-Action (VLA) models genuinely understand and act upon natural language instructions, we designed a **goal replacement** evaluation. In this task, the target object specified in both the instruction and the task goal was replaced with an alternative object from the same scene, while keeping the rest of the environment unchanged. For example, an

Table 8: Pairwise evaluation results across different perturbation dimensions.

|            | Object | Background | Light | Camera | Robot | Noise |
|------------|--------|------------|-------|--------|-------|-------|
| **Object**     | **71.75** | –          | –     | –      | –     | –     |
| **Background** | 57.00  | **85.75**  | –     | –      | –     | –     |
| **Light**      | 57.20  | 67.10      | **82.10** | –  | –     | –     |
| **Camera**     | 35.95  | 37.70      | 39.65 | **57.30** | –  | –     |
| **Robot**      | 24.40  | 29.95      | 29.65 | 19.05  | **39.10** | – |
| **Noise**      | 44.55  | 51.05      | 54.00 | 36.70  | 22.15 | **71.50** |

original instruction such as *"pick up the alphabet soup and place it in the basket"* could be modified to *"pick up the tomato sauce and place it in the basket"*. We performed this manipulation on the *object* suite, where misalignment between model actions and instructions was most pronounced. Figure 5(b) summarizes the performance drop across models, while the rollout cases in Figure 10 reveal how these degradations manifest in execution.

From these results, we observed two key patterns:

1. **Lack of cross-object generalization in instruction following.** Across all tested instances, models failed to adapt to the new target specified in the instruction, with success rates in replaced-target tasks dropping nearly to zero. This drop was particularly dramatic for OpenVLA-OFT, whose accuracy in the modified target setting diminished from high baseline values to almost complete failure. This confirms that the robustness observed in earlier language perturbation experiments did not originate from true linguistic comprehension—the models appear to ignore linguistic signals and rely instead on fixed, learned perception–action associations.

2. **Over-reliance on fixed vision–action mappings rather than dynamic instruction-based planning.** In nearly all rollout cases (Figure 10), the model performed the original action for the original target even when the instruction had explicitly changed. For example:
   - In case (a), the new instruction specified picking up the *butter*, yet the model still picked up the *alphabet soup* as in the original task.
   - In case (c), the model was instructed to pick up *tomato sauce*, but executed the original *butter* action.
   - Similar behavior was observed in (d) and (e), where the model persisted with the original target (e.g., *chocolate pudding*, *cream cheese*) rather than adjusting to the new goal.

These behavioral patterns indicate that the VLA models in our study function more like "visual pattern matchers" mapping scene configurations to predetermined action sequences, rather than integrating task-relevant

# G   DETAILS OF THE COMPOSITIONAL GENERALIZATION EXPERIMENTS

## G.1   SUCCESS RATE RECORD

Table 8 reports the pairwise success rates across different perturbation dimensions. Each diagonal entry corresponds to the performance under a single perturbation dimension, while the off-diagonal entries represent joint perturbations of two dimensions. This analysis allows us to examine not only the robustness of models to isolated disturbances, but also the interaction effects between multiple perturbations, which are critical for assessing ompositional generalization in realistic robotic scenarios.

## G.2   SIGNIFICANCE EXPERIMENTS FOR COMPOSITIONAL GENERALIZATION

To ensure that the observed deviations between the expected product-based success rates and the actual joint success rates are not due to random chance, we conduct significance experiments. Specifically, we aim to statistically validate whether the negative compositionality gaps indeed reflect

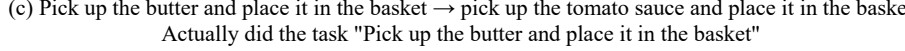

(a) Pick up the alphabet soup and place it in the basket → pick up the butter and place it in the basket
Actually did the task "Pick up the alphabet soup and place it in the basket"

(b) Pick up the alphabet soup and place it in the basket → pick up the butter and place it in the basket
Actually did the task "Pick up the alphabet soup and place it in the basket"

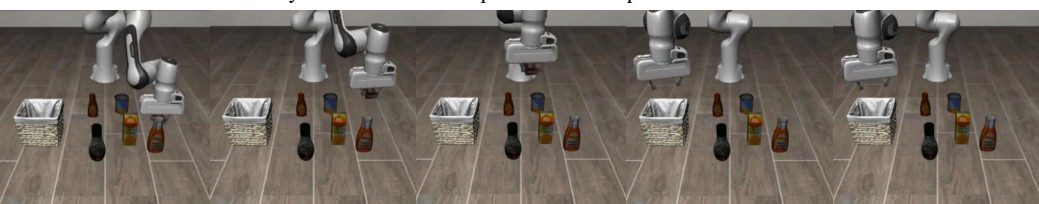

(c) Pick up the butter and place it in the basket → pick up the tomato sauce and place it in the basket
Actually did the task "Pick up the butter and place it in the basket"

(d) Pick up the chocolate pudding and place it in the basket → pick up the salad dressing and place it in the basket
Actually did the task "Pick up the chocolate pudding and place it in the basket"

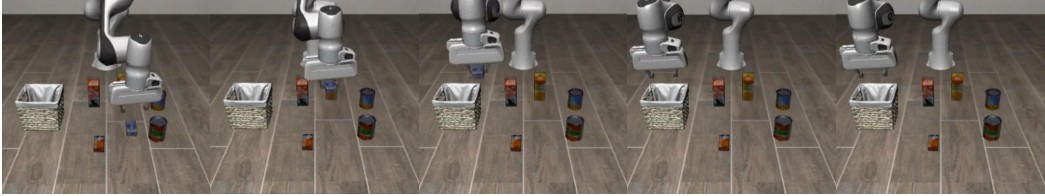

(e) Pick up the cream cheese and place it in the basket → pick up the milk and place it in the basket
Actually did the task "Pick up the cream cheese and place it in the basket"

Figure 10: Behavioral Analysis of Goal Replacement Failures. Case studies showing model responses to modified instructions. For each pair: original→new instruction (above); actually executed behavior (below). The consistent execution of original tasks despite changed targets indicates shallow language processing and strong bias toward memorized visual-action associations.

systematic interaction effects between perturbations, rather than sampling noise arising from finite trials. For this purpose, we adopt the classical chi-square test for independence.

Let $n_{00}$ be the number of samples succeeding under neither of the two perturbations, $n_{01}$ the number succeeding under perturbation 2, $n_{10}$ the number succeeding under perturbation 1, and $n_{11}$ the number succeeding under both perturbations.

- **Chi-square test for independence:** To statistically assess whether the deviation is significant, we consider the $2 \times 2$ contingency table of success counts under perturbations $D_i$ and $D_j$:

| | $D_j = 0$ | $D_j = 1$ | Total |
|---|---|---|---|
| $D_i = 0$ | $n_{00}$ | $n_{01}$ | $n_{0\cdot}$ |
| $D_i = 1$ | $n_{10}$ | $n_{11}$ | $n_{1\cdot}$ |
| Total | $n_{\cdot 0}$ | $n_{\cdot 1}$ | $n$ |

The chi-square statistic is then given by

$$\chi^2 = \sum_{r,c} \frac{(O_{rc} - E_{rc})^2}{E_{rc}},$$

where $O_{rc}$ denotes the observed count and $E_{rc} = \frac{\text{(row total)} \times \text{(column total)}}{n}$ is the expected count under the independence hypothesis.

- **p-value:** Given the chi-square statistic $\chi^2$ and the corresponding degrees of freedom (here $\mathrm{dof} = 1$ for a $2 \times 2$ table), the *p-value* is the probability of observing a test statistic at least as extreme as $\chi^2$ under the null hypothesis of independence:

$$p = P\left(\chi^2_{\mathrm{dof}=1} \geq \chi^2\right),$$

where $\chi^2_{\mathrm{dof}=1}$ denotes a chi-square distribution with 1 degree of freedom. A small $p$-value (e.g., $< 0.05$) indicates strong evidence against the independence assumption.

A large $\chi^2$ value (with small $p$-value) indicates that the joint success/failure distribution under perturbations $d_i$ and $d_j$ deviates significantly from the independence assumption, implying interaction effects between the two perturbations. Conversely, a small $\chi^2$ (large $p$-value) suggests no evidence against independence.

Table 9: Chi-square test results for perturbation pairs

| Perturbation A | Perturbation B | Chi-square | p-value |
|---|---|---|---|
| Object | Env | 4.09 | 4.32e-02 |
| Object | Light | 1.23 | 2.68e-01 |
| Object | Camera | 7.55 | 6.01e-03 |
| Object | Robo init | 6.13 | 1.33e-02 |
| Object | Noise | 9.42 | 2.14e-03 |
| Env | Light | 2.37 | 1.24e-01 |
| Env | Camera | 26.1 | 3.33e-07 |
| Env | Robo init | 4.87 | 2.74e-02 |
| Env | Noise | 16.1 | 6.07e-05 |
| Light | Camera | 12.1 | 4.92e-04 |
| Light | Robo init | 2.79 | 9.48e-02 |
| Light | Noise | 4.53 | 3.34e-02 |
| Camera | Robo init | 6.76 | 9.31e-03 |
| Camera | Noise | 5.51 | 1.90e-02 |
| Robo init | Noise | 14.3 | 1.59e-04 |

From Table 9, it can be observed that most perturbation pairs yield large $\chi^2$ values, with correspondingly tiny $p$-values, below conventional significance thresholds (0.05). This indicates that the joint distribution under different perturbations deviates strongly from the independence assumption, implying clear interaction effects between perturbations.

Overall, the results consistently demonstrate that perturbation interactions are significant and cannot be ignored when evaluating compositional generalization.

## H   FAILURE CASES STUDY

To gain deeper insights into the model's failure mechanisms beyond aggregate performance metrics, we conduct a qualitative analysis of characteristic error patterns across different perturbation

types. This case study reveals how each perturbation dimension induces distinct failure modes in object localization, task understanding, and action execution, providing explanatory context for the quantitative results presented in previous sections. Typical failure cases can be seen in Figures 17 to 19.

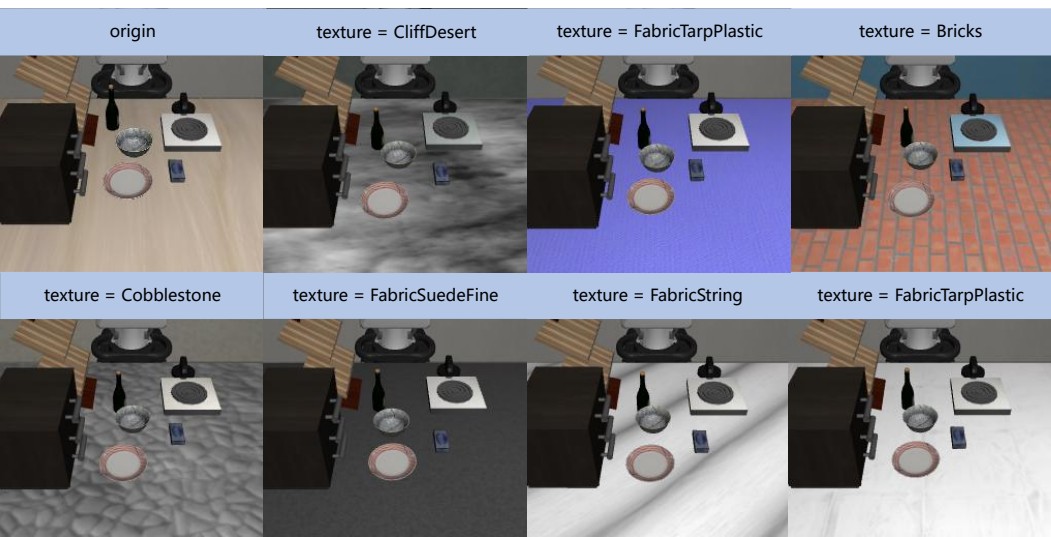

Figure 11: Rendering results with background texture perturbations. The top-left image is the original; the others show results with the textures as labeled.

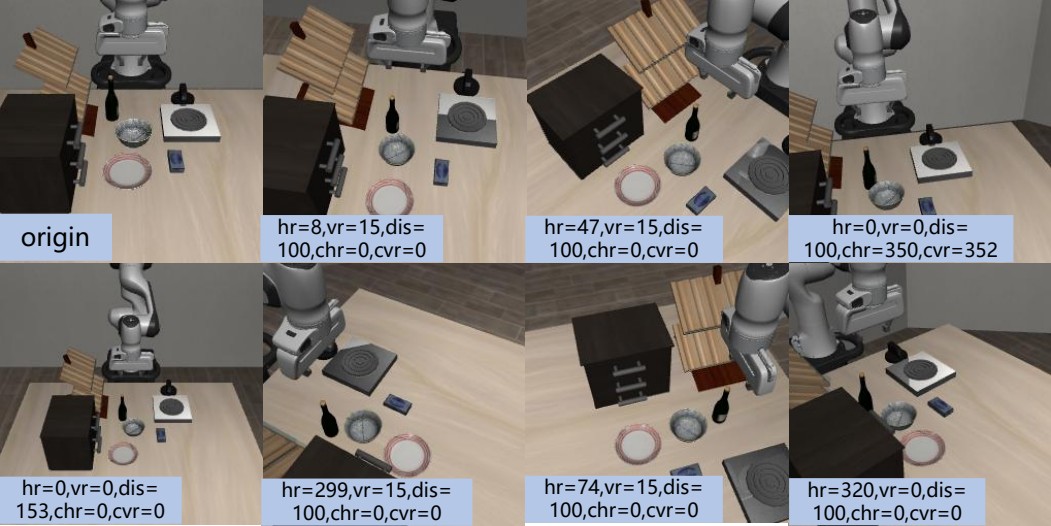

Figure 12: Rendering results under background texture perturbations, comparing the original image (top-left) with transformed versions. The labels denote the following transformation parameters: hr (horizontal rotation angle), vr (vertical rotation angle), dis (distance pulled away), chr (in-place horizontal rotation angle), and cvr (in-place vertical rotation angle).

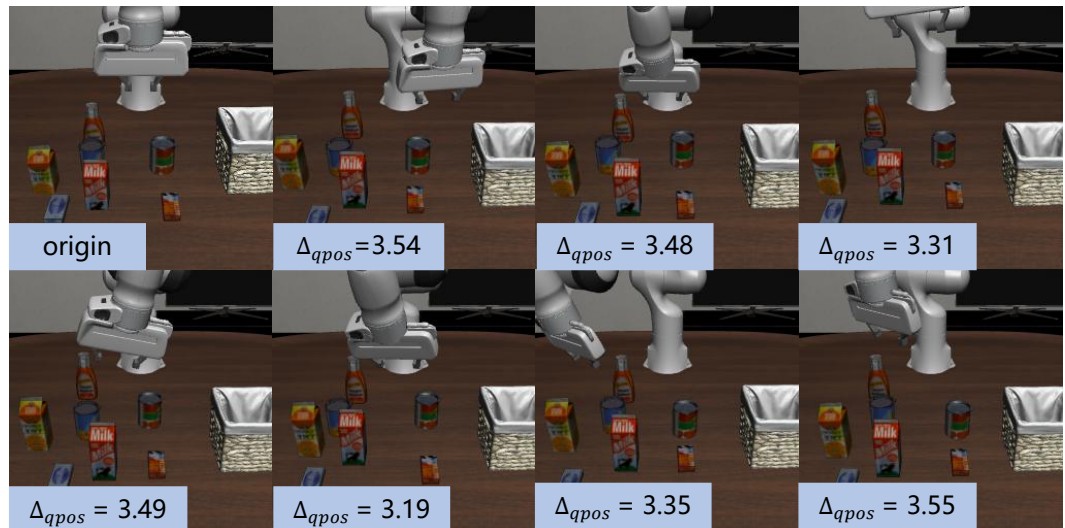

Figure 13: Rendering results with robot initial state perturbations. The top-left image is the original; the others show results with the norm of the change in the robot's joint angles as labeled.

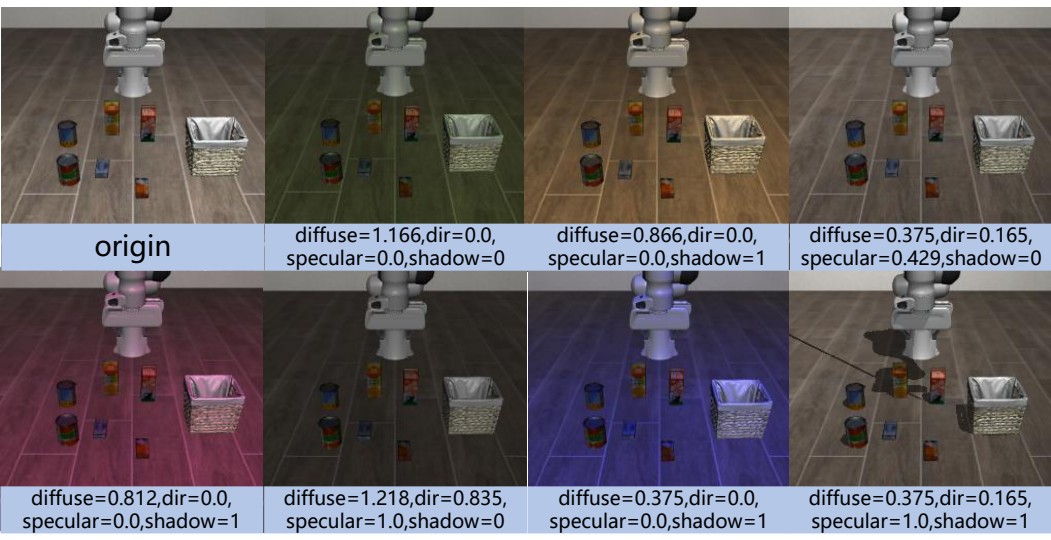

Figure 14: Rendering results with light perturbations. The top-left image is the original; the others show results with the relative change as labeled.

## I    DETAILED RESULTS OF LIBERO-PRO

This section presents a comprehensive analysis of generalization performance under diverse perturbations on the LIBERO-Pro benchmark. Table 10 provides detailed success rates across seven perturbation categories (Camera, Robot Initialization, Language Instruction, Lighting, Background, Sensor Noise, and Scene Layout) for various VLA methods, with results further broken down by task suite (Spatial, Object, Goal, and Long). The comparative analysis reveals significant differences in robustness patterns across methods and perturbation types, offering valuable insights for understanding model generalization capabilities.

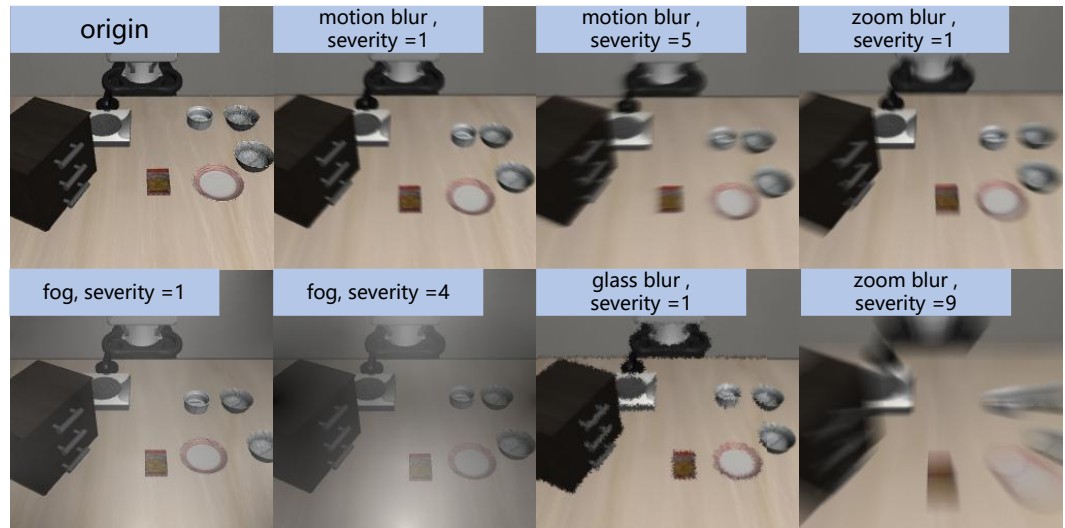

Figure 15: Rendering results with sensor noise perturbations. The top-left image is the original; the others show results corresponding to the type and severity of the applied noise, as indicated by the labels.

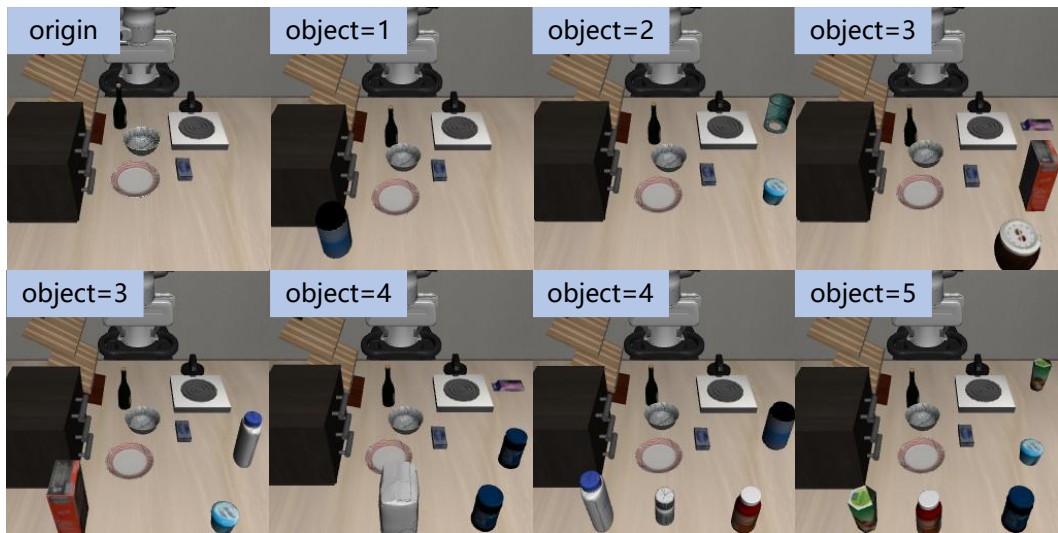

Figure 16: Rendering results with object layout perturbations. The top-left image is the original; the others show results with the number of added objects as labeled.

Table 10: Detailed generalization performance comparison across different perturbation types on the LIBERO-pro benchmark. The table reports success rates (%) for various VLA methods under seven distinct perturbation categories and their average (Total). Results are further broken down by task suite to provide fine-grained insights into each method's robustness capabilities.

| | Camera | Robot | Language | Light | Background | Noise | Layout | Total |
|---|---|---|---|---|---|---|---|---|
| OpenVLA | | | | | | | | |
| Spatial | 0.0 | 3.7 | 27.7 | 12.3 | 50.4 | 12.0 | 40.7 | 19.4 |
| Object | 0.5 | 4.5 | 21.0 | 1.0 | 45.2 | 11.4 | 22.4 | 14.0 |
| Goal | 2.5 | 2.7 | 21.5 | 9.0 | 27.1 | 19.5 | 25.6 | 15.1 |

Table 10 (continued)

| | Camera | Robot | Language | Light | Background | Noise | Layout | Total |
|---|---|---|---|---|---|---|---|---|
| Long | 0.0 | 3.0 | 22.2 | 10.6 | 19.4 | 17.6 | 28.3 | 14.3 |
| Avg | 0.8 | 3.5 | 23.0 | 8.1 | 34.8 | 15.2 | 28.5 | 15.6 |
| **OpenVLA-OFT** | | | | | | | | |
| Spatial | 88.3 | 40.0 | 80.5 | 98.3 | 97.3 | 96.3 | 93.9 | 84.0 |
| Object | 38.9 | 25.4 | 99.0 | 73.7 | 97.6 | 72.3 | 71.8 | 66.5 |
| Goal | 62.0 | 25.2 | 53.2 | 93.9 | 92.5 | 75.2 | 59.1 | 63.0 |
| Long | 38.7 | 38.2 | 87.0 | 89.4 | 86.8 | 63.5 | 76.9 | 66.4 |
| Avg | 56.4 | 31.9 | 79.5 | 88.7 | 93.3 | 75.8 | 74.2 | 69.6 |
| **OpenVLA-OFT_w** | | | | | | | | |
| Spatial | 8.8 | 39.7 | 83.6 | 88.4 | 99.2 | 55.3 | 82.7 | 62.5 |
| Object | 10.1 | 31.4 | 76.4 | 85.9 | 96.4 | 48.3 | 66.3 | 56.0 |
| Goal | 16.4 | 39.9 | 47.1 | 85.3 | 89.0 | 54.9 | 61.8 | 53.3 |
| Long | 6.2 | 43.8 | 77.3 | 46.0 | 90.7 | 43.0 | 72.0 | 52.2 |
| Avg | 10.4 | 38.6 | 70.5 | 76.8 | 93.6 | 49.9 | 69.9 | 55.8 |
| **OpenVLA-OFT_m** | | | | | | | | |
| Spatial | 55.3 | 19.7 | 92.7 | 100.0 | 92.3 | 85.2 | 94.5 | 75.4 |
| Object | 70.2 | 18.1 | 98.5 | 100.0 | 91.9 | 94.1 | 77.4 | 77.1 |
| Goal | 56.6 | 17.1 | 47.6 | 87.8 | 94.7 | 65.7 | 46.6 | 56.2 |
| Long | 41.0 | 31.8 | 88.3 | 82.1 | 85.5 | 69.9 | 61.0 | 63.9 |
| Avg | 55.6 | 21.7 | 81.0 | 92.7 | 91.0 | 78.6 | 68.7 | 67.9 |
| **NORA** | | | | | | | | |
| Spatial | 4.3 | 50.9 | 63.8 | 66.8 | 65.5 | 12.5 | 84.6 | 47.6 |
| Object | 0.5 | 28.4 | 76.4 | 25.3 | 54.8 | 5.7 | 55.8 | 34.4 |
| Goal | 2.9 | 31.1 | 56.6 | 60.6 | 60.5 | 18.2 | 53.9 | 38.8 |
| Long | 1.2 | 39.4 | 64.0 | 30.3 | 54.0 | 15.1 | 59.5 | 36.3 |
| Avg | 2.2 | 37.0 | 65.1 | 45.7 | 58.6 | 12.8 | 62.1 | 39.0 |
| **WorldVLA** | | | | | | | | |
| Spatial | 0.0 | 44.3 | 46.3 | 65.1 | 19.8 | 11.7 | 46.1 | 32.5 |
| Object | 0.0 | 26.4 | 57.2 | 20.5 | 17.3 | 18.0 | 53.6 | 28.6 |
| Goal | 0.3 | 30.6 | 42.2 | 68.8 | 30.3 | 13.5 | 47.4 | 31.8 |
| Long | 0.0 | 12.2 | 20.6 | 20.4 | 1.7 | 1.6 | 4.4 | 8.2 |
| Avg | 0.1 | 27.9 | 41.6 | 43.7 | 17.1 | 10.9 | 38.0 | 25.0 |
| **UniVLA** | | | | | | | | |
| Spatial | 1.1 | 52.6 | 83.9 | 96.6 | 90.7 | 15.7 | 69.5 | 55.5 |
| Object | 0.0 | 42.2 | 86.9 | 25.6 | 81.5 | 10.4 | 27.3 | 36.7 |
| Goal | 3.9 | 37.9 | 45.6 | 89.6 | 78.3 | 33.5 | 22.6 | 40.7 |
| Long | 1.9 | 53.2 | 64.2 | 65.7 | 74.4 | 25.4 | 16.4 | 39.9 |
| Avg | 1.8 | 46.2 | 69.5 | 69.0 | 81.0 | 79.0 | 31.9 | 52.1 |
| **pi0** | | | | | | | | |
| Spatial | 17.8 | 6.6 | 58.8 | 89.7 | 90.7 | 90.9 | 89.1 | 60.7 |
| Object | 22.2 | 8.3 | 70.0 | 90.9 | 91.1 | 87.0 | 76.2 | 61.4 |
| Goal | 12.3 | 5.6 | 39.3 | 84.2 | 76.5 | 76.5 | 44.7 | 44.9 |
| Long | 3.8 | 3.6 | 68.4 | 74.5 | 69.5 | 64.4 | 69.6 | 48.4 |
| Avg | 13.8 | 6.0 | 58.8 | 85.0 | 81.4 | 79.0 | 68.8 | 53.6 |
| **pi0_Fast** | | | | | | | | |
| Spatial | 87.2 | 26.9 | 84.2 | 37.0 | 97.7 | 93.2 | 95.5 | 74.4 |
| Object | 72.0 | 27.6 | 71.5 | 71.0 | 95.2 | 93.1 | 84.5 | 72.7 |
| Goal | 70.8 | 20.5 | 47.3 | 95.3 | 60.9 | 69.7 | 51.6 | 57.5 |

Table 10 (continued)

|        | Camera | Robot | Language | Light | Background | Noise | Layout | Total |
|--------|--------|-------|----------|-------|------------|-------|--------|-------|
| Long   | 33.2   | 12.0  | 43.6     | 91.6  | 44.6       | 46.1  | 47.8   | 43.4  |
| Avg    | 65.1   | 21.6  | 61.0     | 73.2  | 73.2       | 74.4  | 68.8   | 61.6  |
| *RIPT-VLA* | | | | | | | | |
| Spatial | 85.4  | 38.0  | 99.7     | 99.7  | 100.0      | 92.0  | 92.3   | 85.8  |
| Object  | 37.9  | 26.4  | 80.8     | 85.9  | 99.2       | 68.0  | 70.1   | 64.3  |
| Goal    | 65.7  | 23.2  | 45.4     | 74.2  | 79.7       | 71.0  | 59.8   | 58.0  |
| Long    | 34.1  | 38.4  | 88.3     | 93.4  | 89.3       | 66.4  | 79.2   | 67.5  |
| Avg     | 55.2  | 31.2  | 77.5     | 88.3  | 91.6       | 73.5  | 74.2   | 68.4  |
| **Ours** | | | | | | | | |
| Spatial | 98.4  | 31.7  | 96.0     | 99.3  | 98.8       | 86.3  | 97.8   | 86.1  |
| Object  | 97.0  | 24.6  | 100.0    | 99.7  | 98.8       | 97.4  | 82.8   | 84.5  |
| Goal    | 93.9  | 24.7  | 55.1     | 96.8  | 94.0       | 93.4  | 53.9   | 70.7  |
| Long    | 82.6  | 40.7  | 94.8     | 83.2  | 85.1       | 80.6  | 80.3   | 77.7  |
| Avg     | 92.8  | 30.3  | 85.8     | 94.9  | 93.9       | 89.3  | 77.6   | 79.5  |

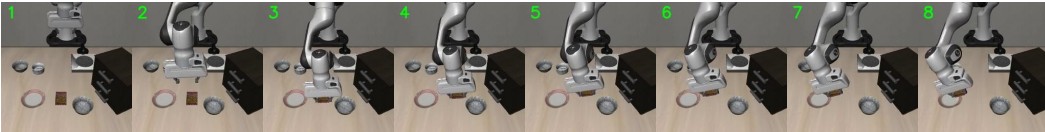

Camera – changes in camera position cause the model to localize the target object inaccurately.

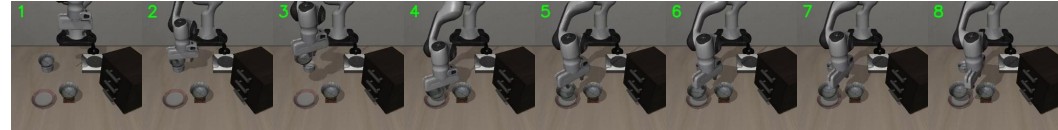

Language - modified language description sets the task object as darkcolored dish, but the model incorrectly localizes cookies

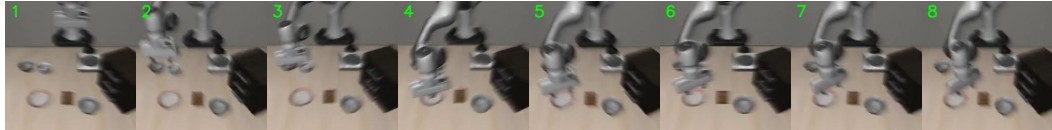

Light - variations in light source position create shadows, leading to biased localization of the target object.

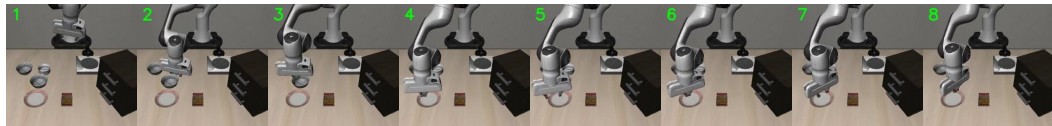

Noise - added noise blurs the image, resulting in inaccurate localization of the target object

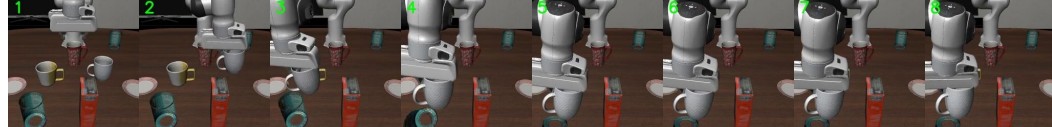

Robot - changes in the robot arm's initial position cause deviations in path planning and final positioning.

Layout - additional distractor objects lead to mislocalization of the target plate, with a nearby object being mistakenly recognized as the plate.

Figure 17: Failure Mode Analysis Across Perturbation Types. Visualization of characteristic failure patterns induced by each perturbation dimension, revealing distinct vulnerability profiles: camera shifts cause viewpoint-dependent localization errors; language modifications lead to semantic misinterpretations; lighting variations introduce shadow artifacts; sensor noise produces feature corruption; initial state changes affect trajectory planning; and object distractors trigger recognition confusion.

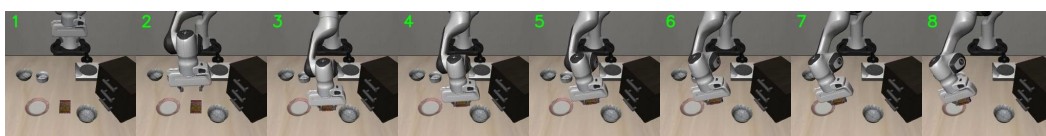

language - modified language description sets the task object as darkcolored dish, but the model incorrectly localizes cookies

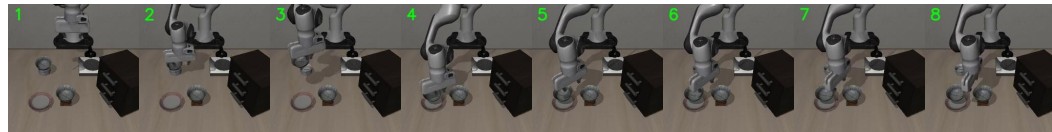

light - variations in light source position create shadows, leading to biased localization of the target object.

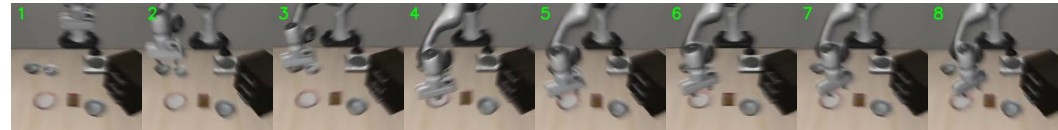

noise - added noise blurs the image, resulting in inaccurate localization of the target object

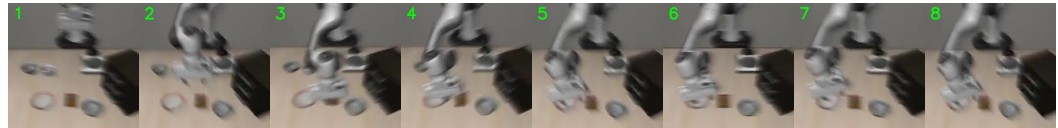

noise - added noise blurs the image, resulting in inaccurate localization of the target object

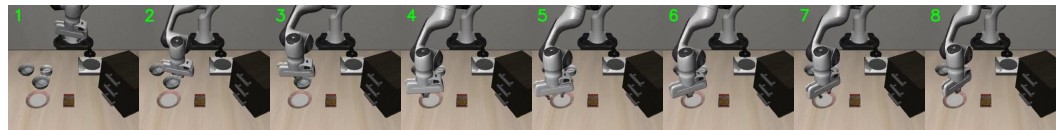

initstate - changes in the robot arm's initial position cause deviations in path planning and final positioning.

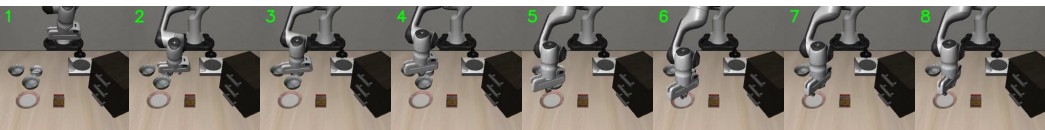

initstate - changes in the robot arm's initial position cause deviations in path planning and final positioning.

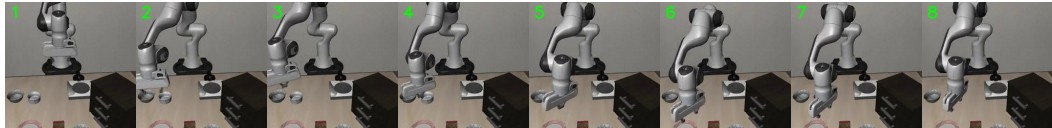

camera – changes in camera position cause the model to localize the target object inaccurately.

Figure 18: Failure Mode Analysis Across Perturbation Types. Visualization of characteristic failure patterns induced by each perturbation dimension, revealing distinct vulnerability profiles: camera shifts cause viewpoint-dependent localization errors; language modifications lead to semantic misinterpretations; lighting variations introduce shadow artifacts; sensor noise produces feature corruption; initial state changes affect trajectory planning; and object distractors trigger recognition confusion.

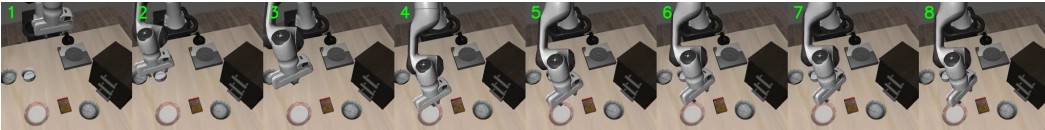

camera – changes in camera position cause the model to localize the target object inaccurately.

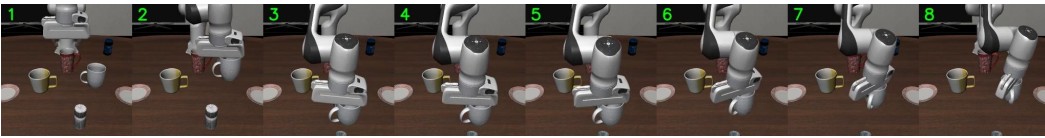

camera – changes in camera position cause the model to localize the target object inaccurately.

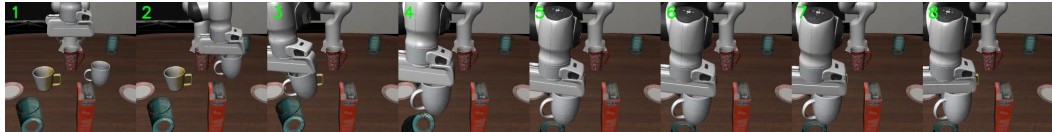

object - additional distractor objects lead to mislocalization of the target plate, with a nearby object being mistakenly recognized as the plate.

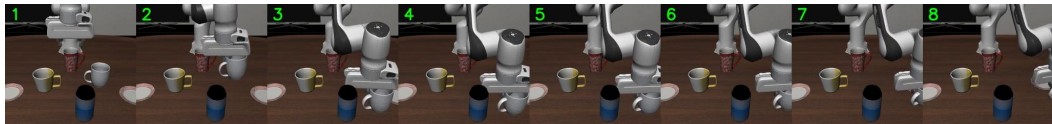

object - additional distractor objects lead to mislocalization of the target plate, with a nearby object being mistakenly recognized as the plate.

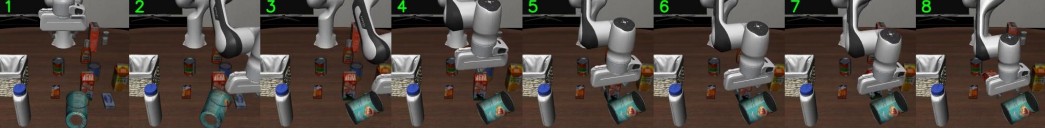

object - additional distractor objects lead to mislocalization of the target plate, with a nearby object being mistakenly recognized as the plate.

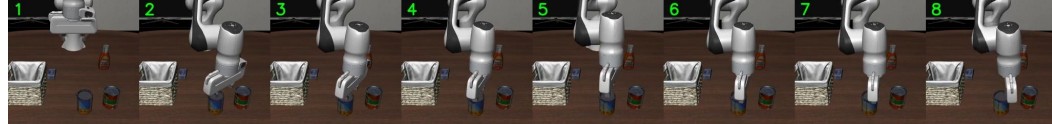

object - The model fails to flexibly rotate the robotic arm, resulting in a collision with a distractor object.

object - after the object position is perturbed, the model fails to correctly localize the object.

Figure 19: Failure Mode Analysis Across Perturbation Types. Visualization of characteristic failure patterns induced by each perturbation dimension, revealing distinct vulnerability profiles: camera shifts cause viewpoint-dependent object localization inaccuracy; object distractors provoke recognition confusion and mislocalization of the target, in some cases leading to incorrect collision-prone trajectories when arm motion flexibility is insufficient.

