# OpenReview forum: "In-depth Robustness Analysis for Vision-Language-Action Models"
_ICLR.cc/2026/Conference — ICLR 2026 Conference Withdrawn Submission_

### Official Review · Reviewer_iq9G · 2025-10-20

**Soundness:** 2
**Presentation:** 3
**Contribution:** 2
**Rating:** 2
**Confidence:** 4

**Summary:**

This paper presents an extensive empirical study, in LIBERO, evaluating the robustness of VLAs across seven axes of variation, e.g., camera viewpoint, lightning, initial state, etc. The authors demonstrate that VLAs are extremely brittle to changes in viewpoint and initial conditions. Moreover, current VLAs have poor language-following capabilities, often ignoring the instruction entirely and relying on other cues to infer task details. To facilitate a more rigorous evaluation, the authors introduce LIBERO-Pro, a new benchmark to assess generalization. They demonstrate that by training on LIBERO-Pro, models can improve robustness by up to 11.5%.

**Strengths:**

First, the quality of the simulation study is high. The authors are commended for evaluating numerous open-source VLAs and performing a systematic analysis on each against a diverse and well-structured set of environmental perturbations.

Second, the paper highlights the poor language-following abilities of current VLAs. While this fact is often apparent to practitioners, it is not often highlighted in academic works. The finding that many VLAs attempt to infer the task from visual cues alone is often an under-appreciated fact. By providing evidence of this short-coming, this paper challenges the community to look beyond success rates and develop more robust methods for policy evaluation.

Introduction of the LIBERO-Pro benchmark is a great addition for the field to bootstrap off of.

**Weaknesses:**

Exclusive reliance on simulation: this work's most fundamental limitation is the lack of real-world demonstrations. The manuscript implicitly makes claims that hold in simulation (LIBERO) and assume they transfer to the real-world, but it is well known that existing simulators are only proxies for real-world deployment. Therefore, it is a significant leap to claim that failure in LIBERO equate to fundamental flaws in the models themselves, especially when many of them are not trained on LIBERO data. This methodological choice weakens the paper's main claims. While a large-scale real-world study is often infeasible, some smaller-scale validation is necessary for the claims to hold.

Limited novelty: The paper's main takeaway, that existing VLAs are brittle to variations in the environment, is a well-established concern within the robot learning community. While the systematic nature of the analysis presented herein is a strength, the novelty is rather limited.

Questionable experimental choices: some specific experimental choices are difficult to interpret. For example, applying a mask to all visual inputs (Figure 2) seems less like a perturbation and more of an entire modality ablation. It is not clear what insight is gained from this beyond confirming that vision is necessary for these tasks.

**Questions:**

Do the authors have an preliminary evidence to suggest that the trends from LIBERO hold in real robot deployment? While simulations can provide initial correlative evidence of downstream performance, its a leap to claim that trends which hold in simulation are guaranteed to transfer. A small-scale real-world study validating even one or two of the key perturbation axes would improve the conclusions.

---

### Official Review · Reviewer_PA3w · 2025-10-30

**Soundness:** 3
**Presentation:** 2
**Contribution:** 2
**Rating:** 2
**Confidence:** 4

**Summary:**

This paper conducts a large-scale empirical study on the robustness of current VLA models under various simulated perturbations. The authors systematically test multiple open VLA baselines (e.g., OpenVLA, SpatialVLA, $\pi$0 variants) across the LIBERO benchmark and introduce a new extension called LIBERO-Pro, claiming to provide a more diverse and challenging evaluation. They further analyze robustness under compositional perturbations and explore how models use (or fail to use) language information.

**Strengths:**

1. The authors clearly put substantial effort into evaluating multiple VLA models under diverse perturbations (lighting, camera, robot initialization, etc.). The results are detailed and reproducible, providing a useful community reference.

2. The finding that many models are insensitive to language perturbations (i.e., rely purely on vision) is interesting and aligns with anecdotal community observations.

3. The attempt to measure performance drop under combined perturbations (as opposed to single-axis changes) is a valuable direction for future diagnostic benchmarks.

**Weaknesses:**

**1. Outdated and limited benchmark choice.**

The entire study is built on LIBERO and its in-house variant, which are low-complexity simulation environments. These benchmarks are already saturated (90–95% success across baselines), meaning further “performance drops” mostly reflect overfitting to the simulator rather than meaningful generalization failures. More modern or realistic benchmarks, e.g., COLOSSEUM [1], RoboTwin [2], would provide stronger evidence.

As it stands, the paper tests “robustness to LIBERO perturbations,” not “robustness to real-world change.”

**2. The most important one: No sim-to-real or physical validation.**

The core motivation is to test robustness, but the entire work remains in simulation. ***The real problem that robot VA/VLA faces is not how high performance we could achieve in the simulation, but its generalization ability in the real world.***
There is no attempt to correlate the measured robustness with real robot performance under even small-scale trials. Without that bridge, it’s totally unclear whether the measured fragilities actually matter for real deployment.

**3. Perturbations lack external validity.**

The perturbations (lighting, background, camera pose) are synthetic and easy to simulate. Real-world degradation, like dirty cameras, HDR shifts, physical calibration drift, deformables, and dynamic occlusion, is far more complex. The current perturbations test algorithmic sensitivity, not embodied robustness.

**4. Over-claiming in the “language is ignored” conclusion.**

The claim that VLAs “ignore language” is overstated. In LIBERO’s single-goal scenes, tasks can often be solved purely visually. The experiments do not isolate cases where language must disambiguate similar visuals. Thus, the evidence supports “conditional language use,” not “complete language neglect.”


**5. Compositionality analysis remains descriptive.**

The Section 5 analysis (pairwise perturbation interactions) produces pretty matrices but provides little mechanistic understanding. It lacks representation-level or causal analysis and no baseline comparisons. The section is more diagnostic than explanatory.

**6. Benchmark extension is useful, but the experiments are self-serving.**

I do think the idea of extending LIBERO toward robustness evaluation has community value: it’s helpful to remind people that existing VLAs are overly benchmark-driven and often brittle under visual or positional shifts. In that sense, creating LIBERO-Pro could have been a good contribution.

However, the way it’s executed makes the experiment somewhat circular. LIBERO itself is already almost “solved”: most recent VLAs hit near-perfect success (over 95%). Now, the authors take that same benchmark, add synthetic “robustness” perturbations, and then pretrain the model exactly on those perturbations. Of course the model performs best on the new benchmark; that outcome is basically guaranteed.

So while the intention is good, the result feels self-fulfilling rather than genuinely diagnostic. It doesn’t necessarily prove that the model is more robust in general — only that it’s optimized for the very disturbances the authors defined. To make this truly convincing, the paper would need to show transfer to unseen perturbations or to a different real-world setup, rather than succeeding on a benchmark built in its own image.

---

*References:*

[1] THE COLOSSEUM: A Benchmark for Evaluating Generalization for Robotic Manipulation. RSS 2024.

[2] RoboTwin 2.0: A Scalable Data Generator and Benchmark with Strong Domain Randomization for Robust Bimanual Robotic Manipulation. 2025

**Questions:**

I acknowledge the contribution of this work to the robotics community, and I appreciate the effort of this work with extensive experiments. However, there still exist many weaknesses at this stage.

Please see the weaknesses.

---

### Official Review · Reviewer_W94C · 2025-10-31

**Soundness:** 4
**Presentation:** 4
**Contribution:** 3
**Rating:** 6
**Confidence:** 4

**Summary:**

The paper gives a broad overview of the performance of the popular VLA models across a series of well-designed tasks that control either 1-dim of variation or multiple dimensions of variation. The evaluation results bring some important conclusions about the weakness of the generalizability of current VLAs, and thus lead to some potential instructions for improvements.

**Strengths:**

1. The motivation is clear, and the paper is easy to follow.
2. The experiments are good and reveal some substantial problems of current VLAs.
3. The paper provides a nice benchmark LIBERO-Pro that helps evaluate the VLA more comprehensively.

**Weaknesses:**

1. While the paper provides a systematic analysis of the modern VLAs,  the scale of performance degradation in some dimension is a common problem for all VLAs. It would be better if the author could give a deeper analysis of this, for example, is this because the high-level, network structure of these VLAs is similar etc.
2. According to the experiments, the language part plays a less important role than vision part in modern VLAs. However, in Table 2, with IMPROVE GENERALIZATION training strategy, it seems the improvement on Language is actually larger than other vision aspects, such as Light and Background, can the author explain this?

**Questions:**

See above

---

### Official Review · Reviewer_ephY · 2025-10-31

**Soundness:** 3
**Presentation:** 3
**Contribution:** 3
**Rating:** 4
**Confidence:** 4

**Summary:**

This paper provides a robustness analysis of Vision-Language-Action (VLA) models. The paper’s core contributions are:

- An analysis of current VLA models through systematic parameter variation.
- A diagnostic framework for identifying and quantifying the impact of perturbations on model performance.
- An analysis of the potential mismatch between claimed multimodal capabilities and actual understanding.

**Strengths:**

- The paper does an excellent job of robust perturbation analysis of VLA models. This work comprehensively studies disturbances for each of the modalities of VLA models and shows the relative vulnerabilities of state-of-the-art VLA models across modalities.
- The paper is overall well-written, with some issues (see weaknesses) on the key findings.

**Weaknesses:**

- Dependence on a Single Simulation Benchmark: A major weakness is the potential lack of generality—the observed brittleness might be an artifact of LIBERO and may not generalize to real-world systems or other simulation platforms.
- Realistic Perturbations: Some of the perturbations for vision may not be that realistic or may be overly synthetic. In the real world, there might be all kinds of visual problems, i.e. partial obstructions of the camera or loose camera, etc.
- Limited Analysis of Language Findings: Previous work (https://arxiv.org/pdf/2411.18676) has shown that models are sensitive to language variations, so claiming that the models may ignore language instructions completely seems like an incorrect claim.
- Focus on Single-Dimensionality: Almost all real-world scenarios often involve changes across multiple axes of modality, i.e. a different background and different lighting, in addition to changes in task instructions. Conclusions associated with single variable changes are somewhat useful for understanding VLA performance, but such limited perturbations may be incomplete or overly optimistic in comparison with real-world environments.

**Questions:**

- Are the findings specific to the LIBERO dataset or does this generalize to other benchmarks?
- Are the vision-based perturbations really comprehensive or have the authors considered other forms of visual perturbations, as mentioned in the “Weaknesses” section?
- How can the authors claim that VLA models are largely insensitive to language perturbations? Prior work has shown that VLA models (e.g. OpenVLA) are very sensitive to language perturbations.
- What happens if more than two of the axes of modality vary? Can the authors comment on the conclusions in such environments, which may be closer to real-world systems with multi-dimensional disturbances?

---

### Note · Authors · 2025-11-13

**Comment:**

Thank you very much for all reviewers' careful review and valuable comments on our paper. After team discussion, we believe that our current method requires more comprehensive ablation studies to verify the effectiveness and necessity of each module. To ensure the rigor and completeness of our research work, we have decided to withdraw this submission. We will continue to improve our work and look forward to contributing to the community in the future.

**Withdrawal Confirmation:**

I have read and agree with the venue's withdrawal policy on behalf of myself and my co-authors.